# IFN-β therapy rescues dysregulated IFN-stimulated proteins, serum cytokines, and neurotrophic factors in multiple sclerosis: Multiplex analysis of short-term and long-term IFN responses

Lei Li[1,2�], Maya Olcer[1�], Zhe Wang[1,3], Yaerin Song[1], Jeffrey Ke[1], Xuan Feng[1*], Anthony T. Reder[1*]

**1** Department of Neurology, A-205, MC-2030, University of Chicago Medicine, Chicago, Illinois, United States of America, **2** Department of Neurology, The 2nd Affiliated Hospital of Harbin Medical University, Harbin, Heilongjiang Province, PR China, **3** Department of Neurology, 1st Affiliated Hospital of Dalian Medical University, Dalian City, PR China

☯ Co first authors.

\* xfeng@bsd.uchicago.edu (XF); areder@bsd.uchicago.edu (ATR)

## Abstract

### Introduction

Dysfunctional regulatory T cells and subnormal responses to interferon-β disrupt the immune system in multiple sclerosis. We probed dysregulated type I IFN pathways *in vitro* and *in vivo* IFN-β to induce transcription factors, cytokines, and neurotrophic factors.

### Methods

36 MS-relevant serum proteins curated to be relevant to MS were detected with multiplex and IFN activity assays plus western blots of mononuclear cells from 15 partial responders (PR) to IFN-β therapy with exacerbations over five years of treatment, and 12 clinical responders (CR) without exacerbations. Response was measured 0, 4, and 24 hours after injection of 16 million units (MU) of IFN-β (double-dose) and 8 MU (standard dose), in clinically stable PR and CR, and 16 MU IFN-β in paired PR during exacerbations, an IFN-resistant state. IFN-β effects after therapy washout were compared to 15 therapy-naïve stable and 13 active RRMS and 18 healthy controls (HC).

### Results

IFN-β injection corrected subnormal levels of p-S-STAT1 transcription factor and induced antiviral MxA and type I IFNs. IFN-β induced anti-inflammatory IL-4, IL-10, IL-12p40, and TNFRII more strongly in stable PR than CR. Th2 cytokines correlated

**Data availability statement:** All of our raw data is in the Supporting information of this manuscript. MS-associated protein expression of treated (Sup Fig 8) and untreated (Sup Fig 9) patients, classic IFN-associated protein expression (Sup Table 1) and corresponding original western blots (S1 raw images). Additional gene expression data of MS patients is detailed in PMID #31648992 and in NCBI Gene Expression Omnibus (GEO) repository, Accession #GSE138064.

**Funding:** Unrestricted research support was from NMSS RG#4509A, State of IL, Bayer Pharmaceuticals, Harbin Medical University (LL), 1st Affiliated Hospital of Dalian Medical University (WZ), and the Joseph Griffin Endowment. The funders had no role in study design, data collection and analysis, decision to publish, or preparation of the manuscript.

**Competing interests:** The authors have declared that no competing interests exist.

**Abbreviations:** AUC, Area under the curve; BBB, Blood-brain barrier; BMI, Body mass index; $X^2$, Chi square test; CNS, Central nervous system; CR, Complete responder to IFN-β therapy; D, Day; dda, Double-dose (16 MU) IFN-β administered during MS activity/exacerbation; dds, Double-dose IFN-β during clinically stable MS; DNS, Data not shown; EAE, Experimental autoimmune encephalitis; EBV, Epstein-Barr virus; EC, Endothelial cells; EDSS, Extended disability scale score (0 normal, 10 death); FC, Fold change; FDR, False discovery rate; Gd, Gadolinium MRI contrast agent; HC, Healthy controls; IFNAR, IFN-α receptor; IFN, Interferon; IFNGR, IFN-γ receptor; IRF9, Interferon regulatory factor 9; ISG, IFN-stimulated (regulated) gene; ISGF3, ISG factor 3; ISRE, IFN-stimulated response element; MS, Multiple sclerosis; MU, Million units; MxA, Myxovirus A protein (Greek myxo=mucus, slime, snot); na, Not applicable; NMO, Neuromyelitis optica; NS, Not significant; PBMC, Peripheral blood mononuclear cells; PCR, Polymerase chain reaction; PR, Partial responder to IFN-β therapy; p-S-STAT1, Phospho-serine signal transducer and activator of transcription; p-Y-STAT1, Phospho-tyrosine STAT1; RRMS, Relapsing/remitting MS; sds, Single dose (8 MU) IFN-β during

with serum vitamin D levels in CR. During exacerbations, IFN-β injections induced Th1, Th2, and neurotropic proteins. After therapy washout, serum IFN-α/β and pro-inflammatory IL-12p70 levels were lower in stable CR than in PR. In therapy-naïve MS, Th1, Th2, and neurotrophic protein levels were surprisingly subnormal and poorly intercorrelated. Long-term IFN-β therapy elevated serum proteins and brought them to a balanced positively-correlated state, echoing HC.

## Conclusions

IFN-β corrects low serum type I IFN levels, enhances responses to subsequent IFN exposure, induces immunoregulatory and neuroprotective proteins, and balances dysregulated and subnormal serum cytokine levels. Low serum IFN and IFN-β-induced proteins link to better long-term response to IFN-β therapy. Correction of immune disruption suggests new mechanisms for immunopathology and therapy.

## Introduction

Multiple sclerosis is an inflammatory brain disease with dysregulated peripheral immunity. Therapy-naïve MS patients exhibit a preponderance of Th1 over Th2 immune function, excessive expression of costimulatory molecules such as CD80 on B cells, dysfunctional CD8 and CD4 regulatory T cells, and a disrupted type I IFN system [1–4].

Subnormal levels of serum type I interferons (IFN-α and IFN-β) plus poor *in vitro* responses to IFN-β by peripheral blood mononuclear cells (PBMC) implicate a fundamental defect in IFN regulation and signaling in therapy-naïve MS patients [3,5] (Fig 1). This defect can be traced to low levels of the IFN-induced phospho-serine-STAT1 transcription factor in PBMC [3]. Low p-S-STAT1 levels fall further during attacks and progression in therapy-naïve patients, suggesting that IFN signaling is linked to the course of MS [3,6]. Environment further modifies IFN signaling. Virus infections induce type I IFN and a shift to Th1 immunity to combat the virus [7]. Vitamin D enhances p-tyrosine-STAT1 formation and boosts IFN signaling. However, vitamin D also causes a shift from Th1 to Th2 immunity and high vitamin D levels correlate with fewer MS exacerbations [2,7]. Disruption of IFN signaling in MS could affect response to environmental influences and to IFN-β therapy.

Clinically, IFN-β therapy reduces MS exacerbations, cognitive decline, CNS MRI lesions, and progression [1,2]. Five years of therapy curtails MS-related deaths by 50% 20 years later and leads to a normal lifespan [8]. However, subnormal responses to type I IFNs may blunt response to therapy in some patients [2,4–6].

Individual serum cytokines do not reliably predict the course of MS or responses to therapies [9]. A constellation of immune markers is a more accurate indicator of the immune state in MS, before and after therapy. In RRMS for instance, IFN-β therapy corrects expression of 95% of the 8,800 dysregulated genes seen in untreated MS PBMC [4]. These genes modify immune regulatory, antiviral, cytostatic, and reparative biology.

stable MS; SEM, Standard error of the mean; SLE, Systemic lupus erythematosus; SOCS, Suppressor of cytokine signaling; Type I IFN, IFN-α plus IFN-β; U-STAT1, unphosphorylated STAT1; Multiplex protein targets are listed in (S2 Fig).

To probe the dysregulated type I IFN signaling pathway during clinical stability and exacerbations, we injected IFN-β to induce MS-relevant intracellular transcription factors and serum proteins over 48 hours. Because exacerbations are linked to weaker response to IFN-β [3], we administered double-dose (dd) IFN-β during MS flares and during stable periods as a paired control, and further tested paired responses during stable periods with standard-dose (sd) injections.

Our data show that stable clinical responders (CR) who were exacerbation-free over 5 years had very low serum type I IFN levels and had benefit from IFN therapy. Partial responders (PR) to IFN-β therapy, even while clinically stable, had paradoxically higher pre-injection serum type I IFN levels and higher *in vivo* and *in vitro* biomarker responses to IFN-β. Thus, high IFN levels were linked to future MS exacerbations, suggesting that these patients benefit less from IFN therapy. In therapy-naïve MS, levels of individual and class-grouped Th1 and Th2 cytokines were broadly disturbed and poorly correlated. Long-term IFN-β therapy brought them to near-normal balance, reduced Th1 cytokines, and increased serum type I IFN, Th2 cytokines, and neurotrophic factors.

## Materials and methods

### Subjects

27 relapsing/remitting MS patients who had received IFN-β therapy for 7.94 ± 1.18 years were followed over five years (Table 1). 12 clinically stable **complete responders (CR)** to IFN-β therapy had no attacks or progression [4,10]. 15 **partial responders (PR)** were assayed while stable and during a clinical exacerbation that occurred more than three months from the stable sample. No other immune-modifying therapies and no glucocorticoids were used within six months before phlebotomy. Patients with severe IFN side effects or therapy failure could not be followed longitudinally.

18 healthy controls (HC) and 28 therapy-naïve RRMS patients, 15 clinically stable for 6 months and 13 with MS exacerbations (a ≥ 1-point EDSS increase EDSS, or 1 point increase on two EDSS subscales [4]), were age- and sex-matched with the IFN-β-treated patients. Subjects, > 18 years old, were recruited for this >5-year study in clinic. The first subject enrolled May 7, 2008, and the last on August 30, 2012, with continued clinical follow-up after that. All understood and signed written informed consents approved by the University of Chicago IRB #10681A.

### Treatment *in vivo* with standard-dose and double-dose IFN-β therapy

IFN-β injections were used to probe the disruptions in type I IFN signaling in MS [2–6]. This was not a therapeutic trial, but it did compare the clinical course to molecular changes. In this prospective investigation, clinically stable patients, after a planned >60-hour washout of IFN-β therapy, received two injections of a standard IFN-β dose or a single dose of IFN-β (8 MU [250 ug] IFN-β-1b SQ, N = 26; or 44 ug IFN-β-1a SQ, N = 1). During an exacerbation, PR had a double-dose injection the morning after contacting the clinic (S1 Fig). Exacerbations induce an IFN-resistant state [3], so a double dose was used to overcome this resistance. Double doses during stable *vs*. active periods thus compared disease state-dependent effects. Paired single doses during stable periods compared effects of standard therapy.

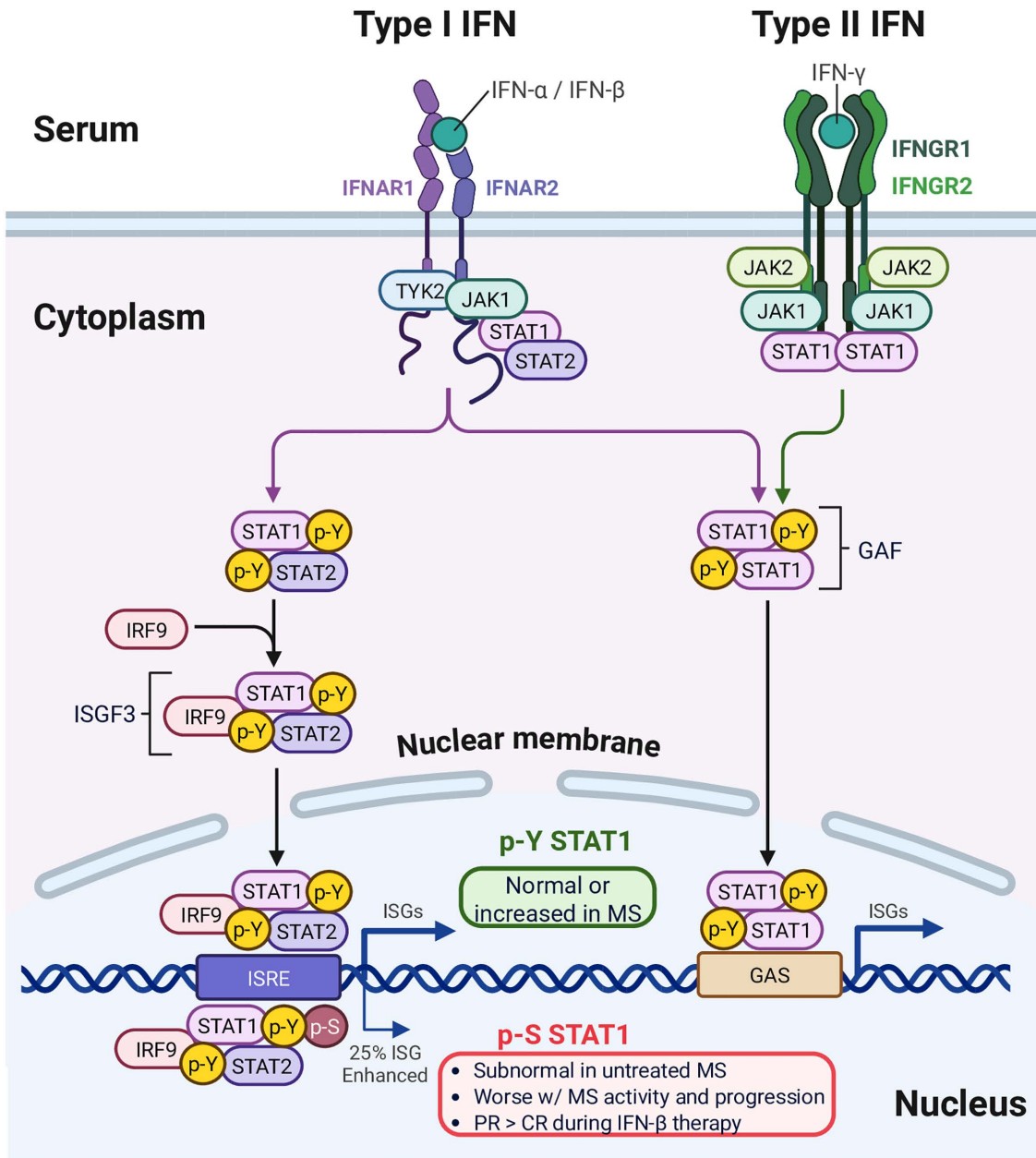

**Fig 1. Interferon signaling pathways are disturbed in MS.**

Blood was drawn at 8 AM in clinic following IFN-β washout in all groups. Patients then self-injected IFN-β and then had carefully-timed phlebotomy at 23.60 ± 0.14 and 47.73 ± 0.11 hours.

Time from last home injection to the study injection during clinically stable periods for single-dose IFN-β was 103.0 ± 6.7 hours, and for double-dose washout was 116.7 ± 9.3 hours (not statistically different), and was equivalent between PR and CR groups. During PR exacerbations, the delay from prior injections was 73.5 ± 9.5 hours (t = 1.76, p < 0.05 *vs*. stable PR). These washout times are well beyond the immediate effects of IFN-β injections. Circulating IFN-β from prior injections is largely eliminated by one hour [11,12]. Type I IFN then induces PBMC mRNA expression of classic IFN-induced genes,

**Table 1. Demographic and clinical characteristics of subjects.**

| Group | A | B | C | D | E | Total |
|---|---|---|---|---|---|---|
| Description | Complete Responder | Partial Responder | Healthy Control | Therapy-naïve, stable | Therapy-naïve, active | |
| # of patients | 12 | 15 | 18 | 15 | 13 | 73 |
| # of samples | 72 | 128 | 18 | 15 | 13 | 246 |
| Stable (S); Active (A) | S, S | S, S, A | na | S | A | na |
| Hours after injection | 0, 4, 24 | 0, 4, 24 | 0 | 0 | 0 | na |
| Hours since injection --t0, washout | 114.5±10.3 | 95.0±9.50 | 0 | 0 | 0 | na |
| Age, years | 48.62±1.59 | 43.33±2.99 | 42.50±4.87 | 45.20±2.64 | 46.33±3.48 | na |
| Race/Ethnic | 8W, 4B | 9W, 6B | 14W, 1H, 1O | 13W, 1H, 1O | 10W, 1B, 1H, 1O | na |
| Sex | 2M, 10F | 3M, 12F | 9M, 9F | 2M, 13F | 13F | na |
| IFN Therapy, years | 8.38±1.78 | 7.52±1.50 | na | na | na | na |
| MS Duration | 14.79±2.83 | 12.51±2.53 | na | 14.00±2.64 | 11.89±2.48 | na |
| EDSS | 2.83±0.58 | 3.17±0.49 | 0 | 3.45±0.52 | 3.11±0.45 | na |

Age and duration of MS between CR, PR, and untreated patients are not significantly different (p>0.05, student's t-test). There were no differences based on race ($X^2$ with Yates' correction). Abbreviations: Active MS/ Exacerbation (A), Stable MS (S); Black (B); East Asian (O), Hispanic (H), White (W); Female (F), Male (M); na=not applicable. Values are Mean±SEM.

including more type I IFN, but this second wave of serum type I IFN is minimal by 40 hours [13,14]. Moreover, post-washout levels of the highly-induced CXCL11/I-TAC protein and the long-lasting MxA marker did not differ between 51–60 hour *vs.* >60-hour washouts, nor between stable *vs.* exacerbating groups ($X^2$ NS).

### *In vitro* activation of intracellular IFN-regulated transcription factors and induction of MxA in PBMC

Peripheral blood mononuclear cells (PBMC) were isolated within 0–4 hours of phlebotomy by density gradient centrifugation (Lympholyte, Cedarlane Labs, Burlington, NC). IFN-induced proteins in PBMC were tested at 0, 1, and 2 days after injections. For restimulation, $5x10^5$ PBMC/ml were incubated with IFN-β-1b at 160 U/ml in 250 ul AIM–V media for 0–90 minutes (STAT1 readouts) or 0–48 hours (MxA) in 96-well round-bottom tissue culture plates at 37°C in 5% $CO_2$ atmosphere.

The activation state of PBMC STAT1 transcription factors, phospho-serine-STAT1 (p-S-STAT1), unphosphorylated STAT1, and induction of intracellular MxA were quantitated with double-antibody Western blots imaged with enhanced chemiluminescence (ECL; GE Health Care, Pittsburgh, PA) [3]. Fresh PBMC were lysed in Laemmli buffer plus protease inhibitors and 1 m*M* Na orthovanadate, Antibodies recognized p-serine-STAT1 (p-S-STAT1), unphosphorylated STAT1 (U-STAT1) (Santa Cruz, Santa Cruz, CA), and MxA (Biogen, Cambridge, MA). p-S-STAT1 forms after p-Y-STAT1 binds DNA (Fig 1). The DNA-bound phosphorylation site of p-S-STAT1 is not detectable by flow cytometry, necessitating quantitation with Western blots [3]. To avoid variability between Westerns, blots were stained for p-STAT1 and actin, then stripped and reprobed for U-STAT1 and actin on the same nitrocellulose membrane. Density units of proteins, as a ratio of STAT/actin or MxA/actin on blots, were quantitated, blinded, with Gene Tool software (Syngene, Frederick, MD). IFN induction of these proteins in HC *vs.* MS and NMO has been detailed in other cohorts [3–7]. We focus here on dissection of *in vitro* and *in vivo* IFN responses in MS.

**Assays of IFN-regulated serum proteins.** 36 serum proteins, type I IFN, and vitamin D were prospectively selected. Targets were based on detectability in serum and on literature showing relevance to MS inflammation, treatment, and CNS function and repair.

**Multiplex protein assay.** We constructed a 29-protein serum multiplex assay (ProcartaPlex™ Immunoassay, Affymetrix/Panomics/eBioscience, Santa Clara, CA) of chemokines, cytokines, adhesion molecules, and neuroendocrine

and neurotrophic proteins relevant to immune control and pathology in MS (S2 Fig). Many serum proteins are not captured by commercial panels, so in preliminary experiments, detectable and MS-relevant proteins potentially regulated by IFN-β were curated from 280 potential targets using diverse multiplex assays (Millipore, RBM, and Panomics) and ELISAs.

Assays of sera stored at -80°C for 0–5 years were all run within a month of each other. We used manufacturer's guidelines with samples in duplicate, but standard curves were expanded from 8 to 9 two-fold dilutions. Patient subtypes and kinetics groups over 0, 24, and 48 hours were randomized between plates to minimize experimental bias.

Three additional serum proteins were measured with other assays. ACTH and CNTF were measured with Milliplex MAP human pituitary magnetic bead panel 1 (Millipore, Billerica, MA). IL-10 levels were confirmed with a high-sensitivity ELISA (Procarta Biosystems, Norwich, UK). Serum IFN and vitamin D are discussed below.

Panel proteins are grouped by their predominant effect on immune cells in the context of MS. For example, BAFF is included with anti-inflammatory cytokines because it enhances Treg function and suppresses experimental autoimmune encephalitis (EAE) [15], and anti-BAFF/BLyS (atacicept) worsens MS [16]. IL-12p40 homodimers inhibit the pro-inflammatory effect of IL-12p70 [17]. MCP-1/CCL2 is a pro-inflammatory chemokine and is neurotoxic in HIV infection. However, CCL2 also induces Th2 cells and polarizes monocytes to M2 cells, and CCL2 levels are low in MS CSF, especially during exacerbations [18,19].

TPO is clustered with anti-inflammatory cytokines because its thymopentin analogue ameliorates EAE by decreasing IL-6 and IFN-γ [20], even though thymopentin increases IL-2 and IFN-γ to inhibit atopic dermatitis [21]. TSLP induces Th2, ILC2, and Treg cells [22]. Neurotrophic BDNF is produced by neurons and Th2 cells and inhibits IL-2 effects [23]. Neurotrophic CNTF has no clear Th1/Th2 effect. CNTF-activated microglia promote neuronal survival and promote oligodendrocyte differentiation, but cause astrocytosis. VEGF-A induces angiogenesis and can disrupt the BBB, but it also activates CD4 Treg and is geroprotective [24]. IL-2, IL7, and IL-15 are predominantly pro-inflammatory and are also homeostatic regulators of immune cell numbers.

**Serum type I IFN activity.** Serum was frozen within four hours of phlebotomy at -80°C. IFN activity was measured at 0, 4, 24, and 48 hours after IFN-β injection. Type I IFN was quantitated with a highly-sensitive mRNA induction assay via real-time PCR of three reporter genes: myxovirus resistance protein (MxA), RNA-dependent protein kinase (PKR), and IFN-induced protein with tetratricopeptide repeats-1 (IFIT-1), all normalized to GAPDH expression [5]. This IFN activity score measures multiple IFN-β and IFN-α subtypes. The 0.1 U/ml limit of detection in this assay is well below the typical 10–20 U/ml threshold of ELISA and eliminates cross-reacting non-IFN proteins seen with ELISA in 28% of sera. Neutralizing Abs (NAb) to IFN-β were measured with an MxA RNA induction assay in A549 lung carcinoma cells [4,25].

**Vitamin D** was quantitated with liquid chromatography/mass spectroscopy (Quest Diagnostics, Schaumburg, IL). Serum values are from the first phlebotomy. Within individuals, repeated vitamin D values closely correlated over time (r = 0.9; data not shown, DNS). Total 25-OH vitamin D levels <20 ng/mL indicate vitamin D deficiency; 20–30 ng/mL suggest insufficiency, 30–100 ng/mL is considered optimal in HC. There was no effect of 16 MU injection of IFN-β-1b on serum vitamin D levels over 48 hours (DNS).

## Statistics

Comparisons between IFN-β dose effects and between active *vs.* stable phases of MS were quantitated with paired t-tests. CR *vs.* PR, treated *vs.* untreated patients, and comparisons with HC, were analyzed with unpaired t-tests. Mann-Whitney U tests were used when multiple values for a protein were below detection threshold. p-values were based on 2-tailed t-tests, except for known IFN-stimulated targets (p-STAT, MxA, and type I IFN) because they had *a priori* expectation of directionality [2–6]. FDR correction was applied to multiple comparisons, a conservative approach, as all pre-defined targets were first curated for relevance to MS. Outliers were assessed with the Interquartile Range (IQR)

method; values that fell <Q1-1.5*IQR or >Q3 + 1.5*IQR were removed. Group analysis used a Wald test for Th1 and Th2 comparisons in heat maps. Values are mean ± SEM. *In vivo* values are given in days; *in vitro* in hours.

## Results

### I. Dose-dependent short-term effects of *in vivo* and *in vitro* IFN-β on PBMC

Demographic characteristics were matched between groups (Table 1). There were more women, as expected in MS, and more black patients than in many MS studies in North America and Europe. There were no obvious sex or race differences in responses.

### A. *In vivo* and *in vitro* activation of p-S-STAT1 transcription factor in PBMC is increased in PR compared to CR

Type I IFNs induce tyrosine phosphorylation of the STAT1 transcription factor. Within 30 minutes, p-Y-STAT1 binds to regulatory DNA elements of IFN-stimulated genes (ISG) (Fig 1). Ten minutes later, additional serine phosphorylation of DNA-bound STAT1 (p-S-STAT1), governs expression of a subset (~25%) of ISG [26]. In untreated MS during exacerbations and disease progression, p-Y-STAT1 levels are normal or slightly increased [3] and were not studied here. In contrast, p-S-STAT1 levels are subnormal in MS PBMC at rest and still low after *in vitro* IFN-β induction in separate MS cohorts, compared to HC [3,5]. We now evaluate IFN-β induction of p-S-STAT1 in clinical subtypes of IFN-treated MS.

#### p-S-STAT1 after IFN-β therapy washout is equivalent in stable and active MS PBMC

After therapy washout, p-S-STAT1 levels in resting PBMC were statistically equivalent between IFN-treated CR (dds), stable PR (dds), and exacerbating PR (dda) (Fig 2A; S1 Table; S11 Fig). In untreated MS in contrast, p-S-STAT1 levels fall during exacerbations [3], suggesting that long-term IFN-β therapy here reverses subnormal p-S-STAT1 levels.

#### In vivo IFN-β activation of p-S-STAT1 in PBMC

In stable PR, double-dose IFN-β injection (dds) elevated p-S-STAT1 levels for 1 and 2 days (Fig 2A, 3-fold relative increase, day 2). In CR, however, there was only a trend for p-S-STAT1 induction (1.3-fold). During exacerbations in PR, p-S-STAT1 increased only at 2 days after double-dose injection (Fig 2C; dda; 2-fold), contrasting with rapid induction at 1 day in stable PR (Fig 2A, B). Single-dose injection (sds) in stable PR and CR had minimal effects at 1–2 days and tended to be less than with double doses (Fig 2B). Thus, stable PR had a persistent response *vs*. CR, and p-S-STAT1 formation was delayed in active PR (Fig 2C).

#### In vitro activation of p-S-STAT1 by IFN-β: Priming is greater in PR than CR

*In vitro* restimulation tested the capacity of PBMC to be activated by additional IFN-β. After IFN washout (day 0), *in vitro* restimulation strongly induced p-S-STAT1 in all groups over 90 minutes (Fig 3A, AUC of stable PR is 3-fold greater than time 0, CR 2-fold, and active PR 2.5-fold, at day 1 or 2). Double-dose injections prolonged the effect of a second stimulation (Fig 3B, 2-day AUC), and induction was delayed in active PR *vs*. stable PR (Fig 3C). Serum NAb to IFN had no observable effect on *in vitro* p-S-STAT1 responses (DNS).

### B. *In vivo* and *in vitro* induction of MxA in PBMC is high in PR compared to CR

Intracellular MxA reflects p-S-STAT1 levels in PBMC and is a biomarker for response to IFN-β therapy [3,6]. After washout, *absolute* MxA protein levels in PR PBMC were higher during exacerbations (4.71 density units at day 0) than during paired stable periods (1.98, p = 0.03) (Fig 4C, left). High MxA levels in PBMC during exacerbations may be triggered by a surge of serum IFN-γ and other cytokines [27].

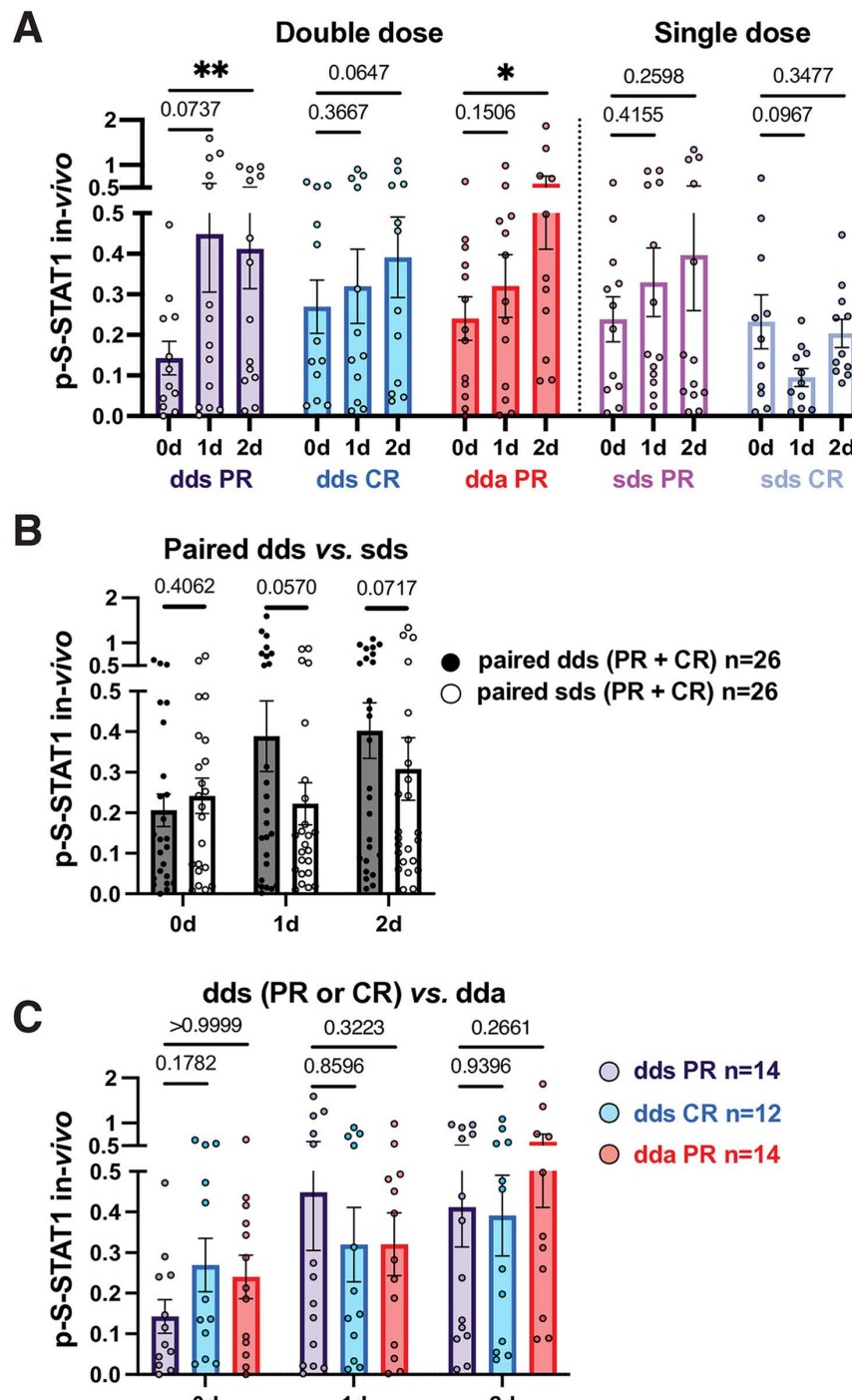

**Fig 2. p-S-STAT1 levels before and after *in vivo* IFN-β induction in PR and CR.**

### In vivo IFN-β induction of intracellular MxA is less in PR than CR

In all stable patients (PR + CR), MxA was elevated 2.12-fold with double-dose injections (2.26 density units on day 0, then 4.71 on day 1) but only 1.28-fold with single-dose at 1 day, persisting at 2 days (Fig 4B). Double-dose injections induced

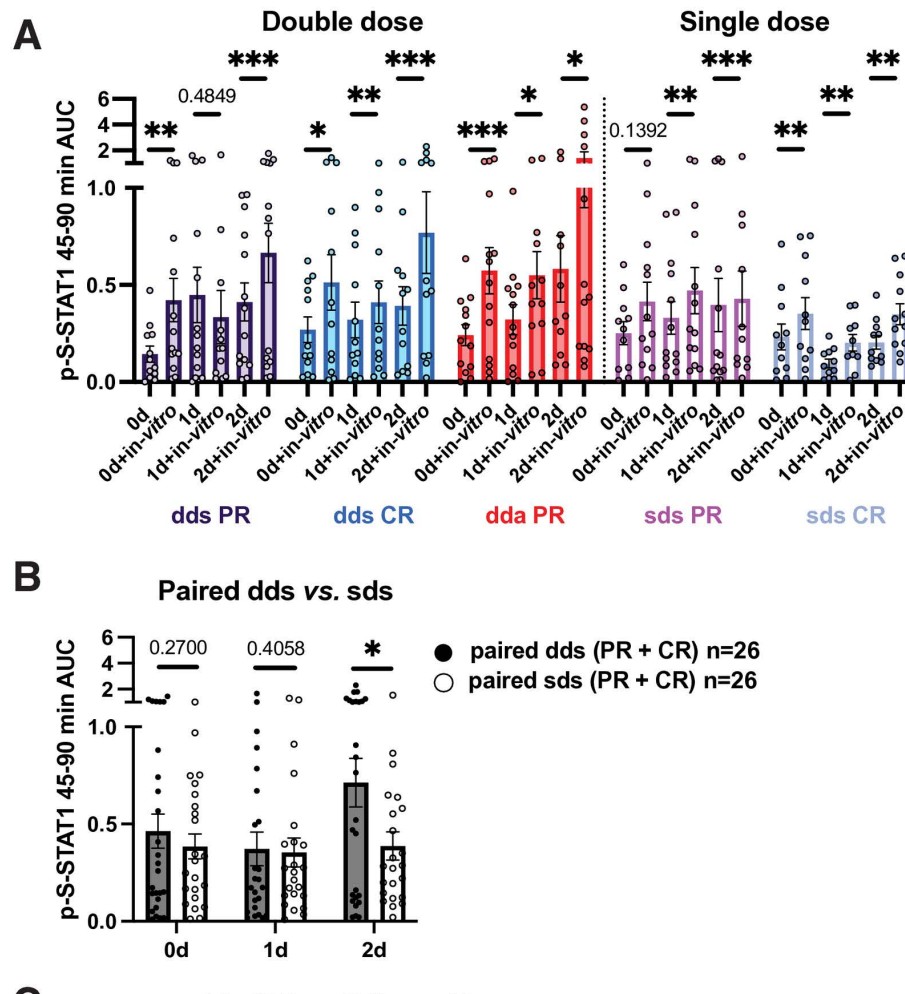

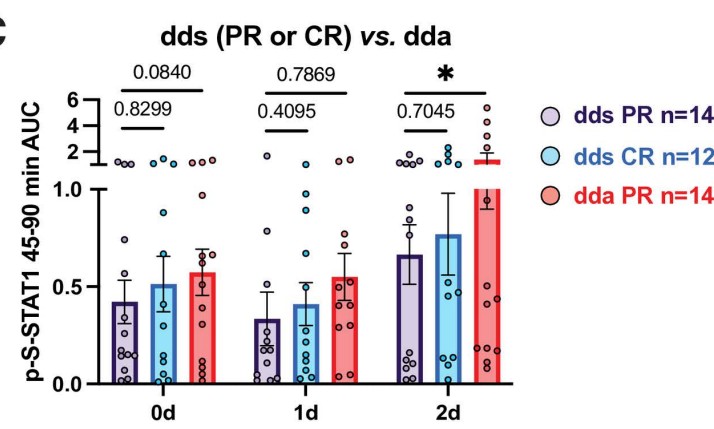

**Fig 3. AUC of p-S-STAT1 levels after *in vitro* IFN-β induction.** The fold change (FC) in p-S-STAT1 after *in vitro* restimulation was enhanced beyond the *in vivo* increase in stable PR, stable CR, and active PR. This "priming" effect from injections amplified responses to additional *in vitro* IFN-β and lasted for at least 4 days after an injection (Fig 3A, C, 0 d washout *vs.* 0 d and then *in vitro*), overcoming the subnormal type I IFN responses that are characteristic of untreated MS ([3], Discussion).

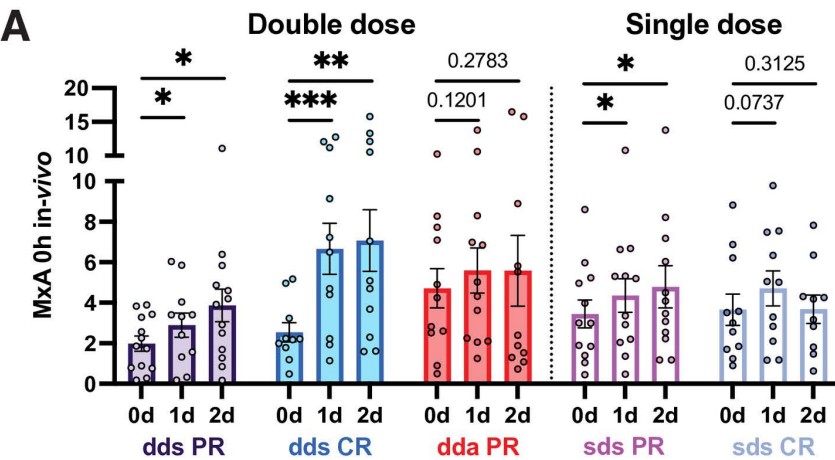

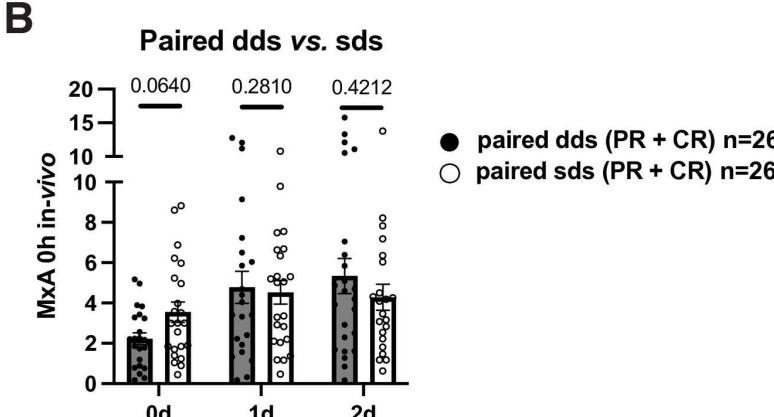

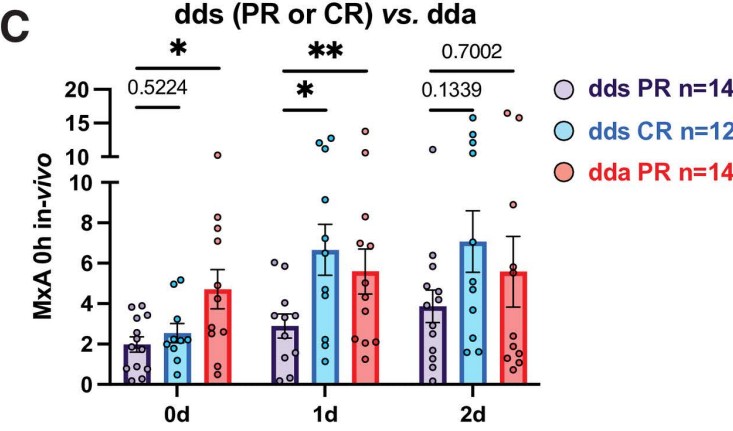

**Fig 4. MxA levels before and after *in vivo* IFN-β induction.**

MxA relatively less in PR at 1 and 2 days (1.5-2-fold) than in CR (3-fold) (Fig 4A). During exacerbations, there was only 1.18-fold relative elevation of MxA at 1 and 2 days (Fig 4A, C) but levels were higher than in stable PR at all times (Fig 4C). Single-dose injections had little effect.

### In vitro induction of MxA by IFN-β: Greater priming in PR than in CR with 8 MU IFN-β

*In vitro* IFN-β restimulation after washout in all stable MS (PR + CR) induced MxA at 0, 1, and 2 days after dd and sd injections: 1.97-fold at 24 hours (Fig 5B; 0d + 24h *in vitro vs.* 0 d washout), 2.18-fold at 48 hours (S3 Fig), and 2.08-fold over 24 + 48 hours (S4 Fig). In stable PR and CR after washout, stimulation *in vitro* increased MxA by 2.03-fold in dds PR (Fig 5C; 1.97 after washout, then 4.01 density units at 24 hours), and 2.01-fold in dds CR. During exacerbations, however, baseline MxA was high and *in vitro* induction (1.27-fold) was lower than in paired stable PR (p = 0.05) (Fig 5C). Thus, after washout, MxA had greater *in vitro* re-induction in stable PR and CR than in active PR, activated, but resistant to restimulation.

### C. *In vivo* and *in vitro* induction of unphosphorylated STAT1 (U-STAT1) in PBMC is elevated in PR compared to CR

IFN-β stimulates synthesis of STAT1 protein. Unphosphorylated STAT1 protein (U-STAT1) provides substrate for more p-STAT1 formation and also induces a subset of ISG, but U-STAT1 induction is not as variable as p-STAT1 activation [28].

### In vivo IFN-β induction of intracellular U-STAT1 is delayed compared to p-S-STAT1 and MxA

After IFN-β washout, stable and active PR had higher U-STAT1 levels than CR (Fig 6A). This may reflect a difference in the IFN signaling set point between the two clinical groups.

After double-dose IFN-β injections, U-STAT1 was induced in stable PR and CR at 2 days more than at 1 day (Fig 6A). p-S-STAT1 and MxA induction, in contrast, is maximal at 1 day. During exacerbations, U-STAT1 was not induced (Fig 6A, C), demonstrating a disease state-dependent lack of response to IFN-β injection. Single-dose injections caused minimal induction in PR and CR.

p-S-STAT1 levels were less than U-STAT1 (Figs 2,3,6). The ratio was lower in stable and active PR (10–30%) than in CR (30–55%), perhaps affecting immune balance and IFN-β responses.

### In vitro induction of U-STAT1 in PBMC is greater in PR than in CR

*In vitro* IFN restimulation induced U-STAT1 in all groups, but more in stable PR than stable CR at 24 and 48 hours, based on absolute values (Fig 7, S5 and S6 Figs). Restimulation had relatively more effect in stable PR than in exacerbating PR at 24 and 48 hours). The lower response during active MS suggests resistance to IFN stimulation [3] and that the priming effect of IFN-β injections is diminished during exacerbations.

Overall, IFN-induced targets were increased after injections and tended to be greater in PR than CR for P-S-STAT1 and U-STAT1 at both IFN doses, and MxA at single doses. We next studied serum type I IFN levels, which regulate the set point of IFN signaling.

### D. Serum type I IFN activity before and after IFN-β injection is elevated in PR compared to CR

### Serum type I IFN levels after therapy washout are greater in PR than in CR

Serum type I IFN after washout in all clinically stable PR was 5.22 ± 2.36 U of IFN activity. In all CR, serum IFN was only 0.29 ± 0.12 U, an 18.1-fold difference (Fig 8, note log scale). Excluding the 4 NAb+ patients (3 PR, 1 CR), values tended to separate even more at time 0, with stable PR at 9.28 ± 5.89 and CR at 0.25 ± 0.17, a 37-fold difference (NS). During

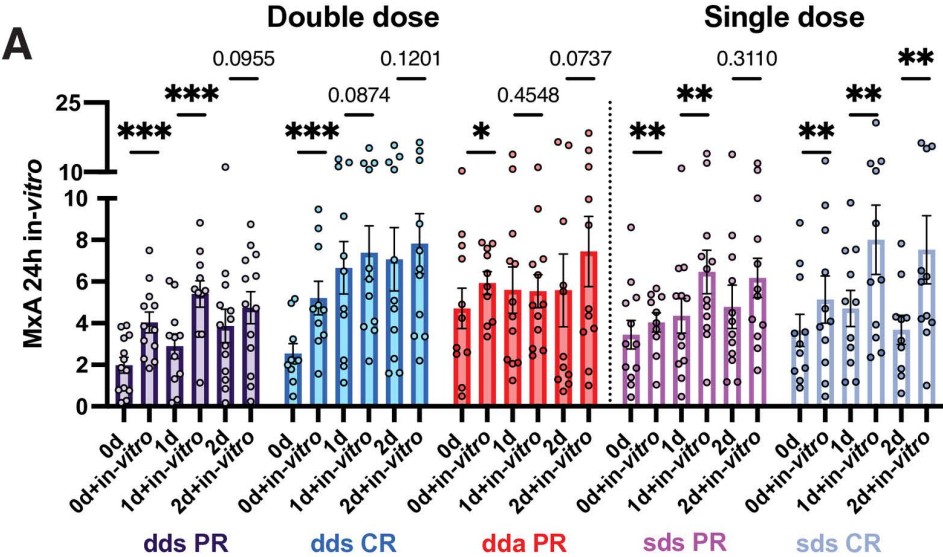

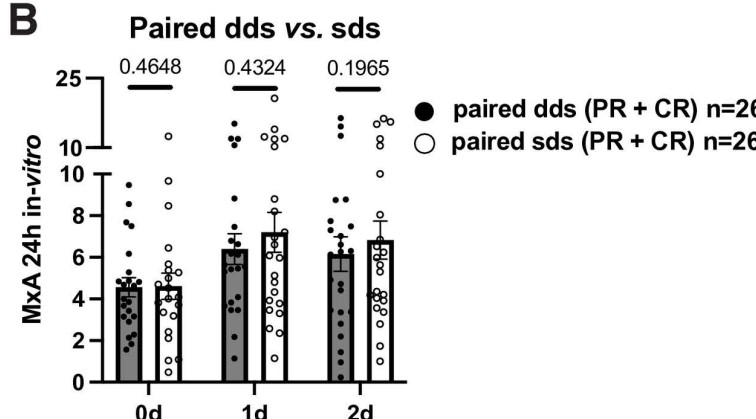

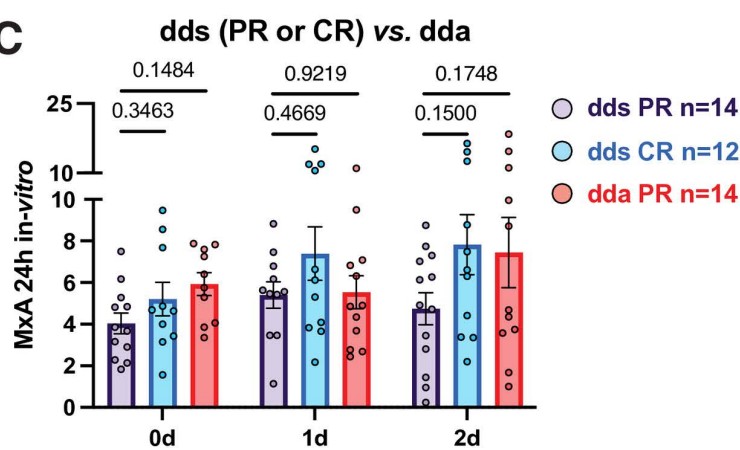

**Fig 5. MxA levels in PBMC 24 hours after *in vitro* IFN-β induction.** Absolute levels after *in vitro* induction were greater 1 and 2 days after injections than after washout (Fig 5A). Thus, during standard IFN-β therapy, the injection "primes" PBMC to respond more vigorously to a second IFN exposure 1 or 2 days later. Although p-S-STAT1 induction was greater in PR with double and single IFN doses (Fig 2), MxA induction was not clearly different between PR and CR with double doses (Fig 4) but may arise from induction of negative regulators such as SOCS.

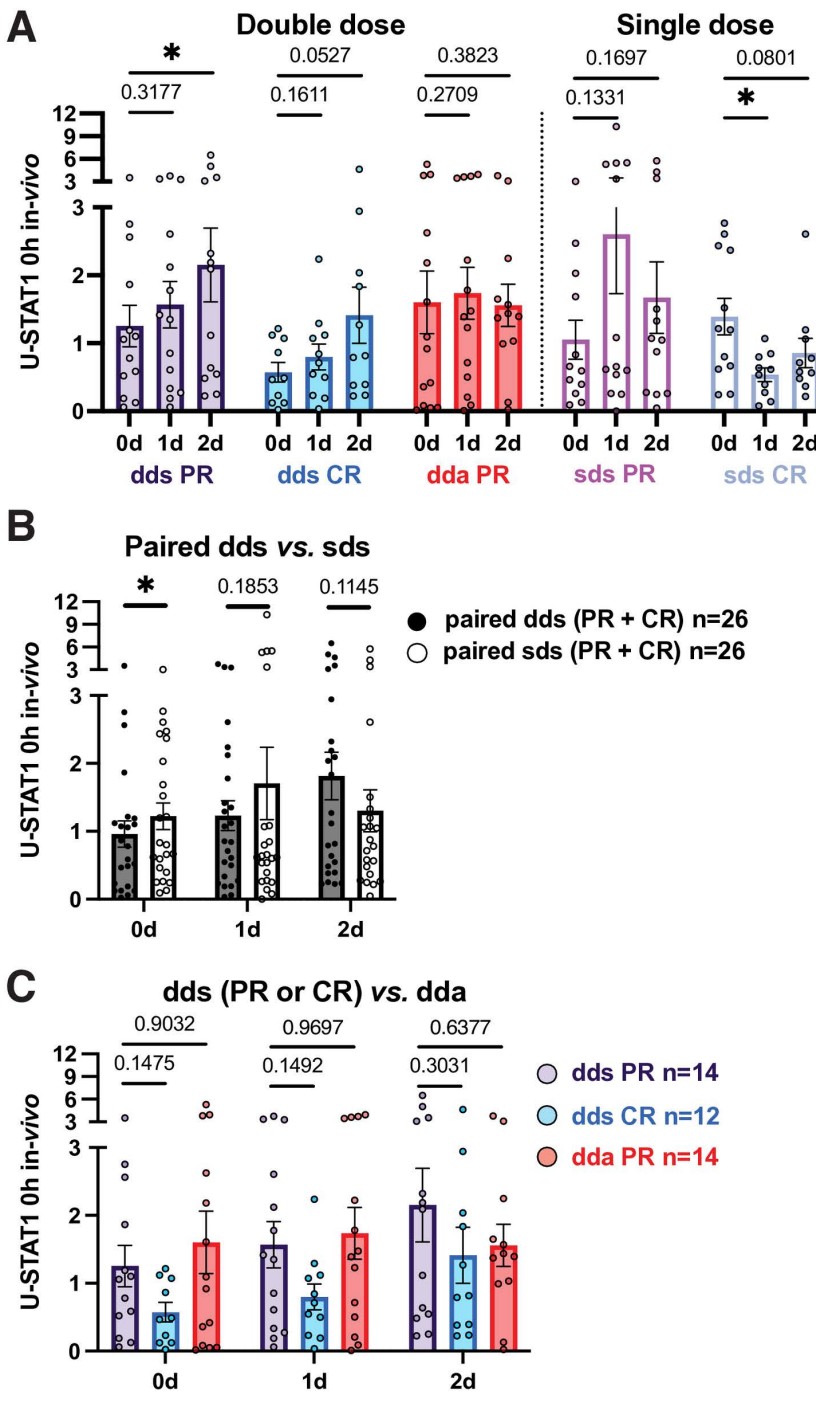

**Fig 6. U-STAT1 levels in PBMC before and after *in vivo* IFN-β induction.**

exacerbations in PR, serum type I IFN activity was intermediate (1.36±0.43). This was 4.7-fold higher than in stable CR (0.29±0.12 U, p=0.03, unpaired t-test), and 3.8-fold lower than in post-washout stable PR (5.22±2.36; 0.006, paired t-test). Values parallel those of IFN-induced transcription factors and MxA, above.

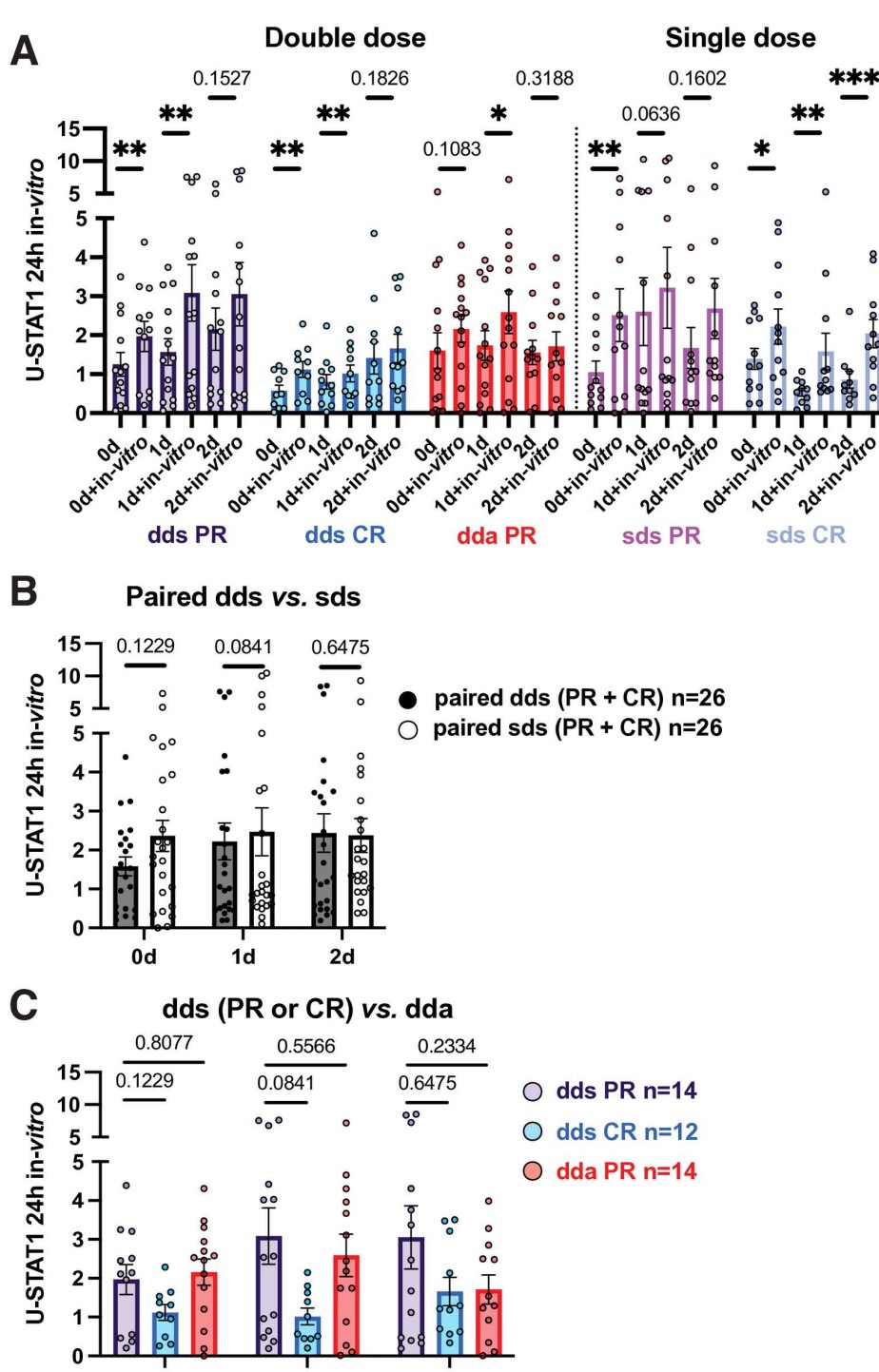

**Fig 7. U-STAT1 levels in PBMC 24 hours after *in vitro* IFN-β induction.**

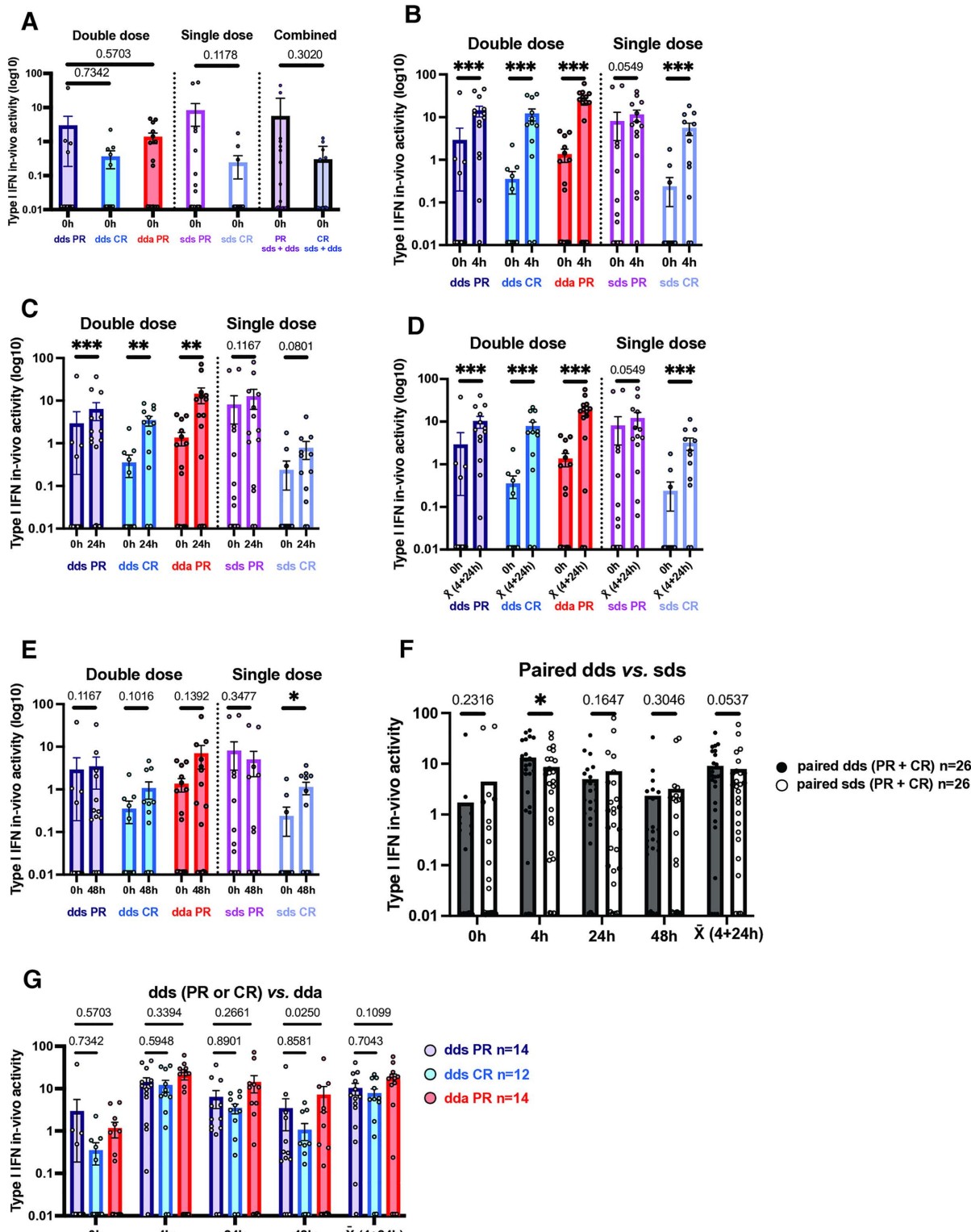

**Fig 8. Serum type I IFN levels after washout and after IFN-β injections.**

Post-washout serum type I IFN correlated with age (r = 0.33, p = 0.01, N = 68). MS is a disease with low type I IFN levels, seen in 150 HC and 47 RRMS-s [5]. This is the converse of SLE, a high-IFN state. In SLE, younger subjects have high serum type I IFN compared to older subjects (r = −0.20, p = 0.001, N = 315) [29]. In both diseases, abnormal IFN levels predominate in younger cohorts – low in MS and high in SLE.

### In vivo IFN-β induction of serum type I IFN is greater in PR than in CR

Injected IFN-β has a half-life of less than one hour, but it rapidly induces more IFN-β, plus multiple subtypes of IFN-α; all then decline over 24 hours [13]. Here, serum type I IFN activity was maximal 4 hours after injections and returned to baseline by 48 hours (Fig 8A–E).

Double-dose IFN-β injections induced more serum type I IFN than single-doses at 4 hours (Fig 8B, F), but both doses were equivalent in PR at 24 and 48 hours (Fig 8C, E). The absolute increase was higher in stable PR than CR with both doses, although the relative increase was higher in CR because of their low post-washout values. During exacerbations, PR started from a lower level of serum IFN (1.36 units of type I IFN activity) before injection than in paired dds PR (2.86 units). In active PR, relative and even absolute induction by injections was increased vs. stable PR at 4, 24, and 48 hours (Fig 8G), paralleling activation of p-S-STAT1 (Fig 3) and cytokines (below).

Post-washout baseline and post-injection serum IFN levels were highly correlated (Fig 8; footnote), suggesting that a high basal level of IFNs primed stronger subsequent responses. dds and sds post-injection type I IFN levels strongly correlated with in vivo-induced IFN at 4, 24, and 48 hours in all stable PR but not in CR, where time 0 IFN levels were often undetectable.

Thus, type I IFN levels after washout and after induction by short-term IFN stimulation were higher in PR than in CR. Post-washout CR had very low serum type I IFN levels. After 8 and 16 MU IFN-β injections, PR have consistently higher serum type I IFN levels, as the absolute rise in IFN-α/β levels is similar in PR and CR. The relative increase is greater in CR due to their low endogenous IFN levels. During exacerbations, IFN-β-induced responses were robust, perhaps resulting from immune factors induced during MS exacerbations.

After 16 MU injection in stable PR, the relative rise in serum type I IFN levels from time 0 was 4.90-fold at 4 hours, 2.16x at 24 hours, and 1.18x at 48 hours (Fig 8B–D). In CR, the relative rise was 34.6-fold at 4 hours, 9.91x at 24 hours, and 3.05x at 48 hours. The relative rise in serum type I IFN levels in exacerbating PR was 18.4-fold at 4 hours, 14.4x at 24 hours, and 5.31x at 48 hours. The absolute increase after injection was greater in dda than dds at all times (23.7 density units in dda vs. 11.2 dds at 4 hours, 18.2 vs. 3.32 at 24 hours, and 5.86 vs. 0.52 at 48 hours).

The higher levels of serum type I IFN, an ISG itself, parallels induction of intracellular p-S-STAT1, and U-STAT1 and is greater in PR than in CR.

### E. An IFN-β injection induces a shift from Th1 to Th2 cytokines in stable patients and induces serum neurotrophic factors during exacerbations

IFN-β induction of serum proteins differed between stable PR and stable CR and was thus linked to future clinical activity. These post-IFN washout serum proteins are contrasted with levels in untreated MS and HC in Results, Section II, A, below.

#### 1. Chemokines: Dose-dependent induction after IFN-β injection

Many chemokines are rapidly induced by type I IFNs [17]. 16 MU IFN-β injection in clinically stable PR and CR induced Th1 cell attractants (CXCL10/IP-10 and CCL2/MCP-1, PR only) and Th2 attractant (CXCL11/I-TAC) at 1 day compared to post-washout time 0 (Fig 9; absolute values in S2 Table). CXCL10, the most strongly induced chemokine, remained elevated at 2 days. During exacerbations, 16 MU IFN-β induced CXCL10 and CXCL11 at 1 day. 8 MU IFN-β injection in

all stable patients induced serum CXCL10 and CXCL11 at 1 and 2 days in all groups (S7 Fig), but less than after 16 MU injections. Thus, chemokine induction was roughly equivalent between stable PR, active PR, and stable CR.

## 2. Cytokines: IFN-β induces Th2 cytokines and suppresses Th1 cytokines in PR and CR

Double-dose IFN-β in all stable MS induced anti-inflammatory IL-9, IL-12p40, TNFRII, and TPO, at 1 day, and diminished pro-inflammatory IL-7 at 2 days (DNS). In stable PR, 16 MU injections tended to induce Th1 cytokines at 1 day, and anti-inflammatory IL-12p40, plus trends for IL-4, IL-10, TNFRII, and TPO, at 1 and 2 days (Fig 9). During exacerbations, IFN-β tended to induceTh2 cytokines, but also pro-inflammatory cytokines. In CR, double doses had a minimal effect on Th1 and Th2 cytokines in all groups (S7 Fig).

Thus, double-dose IFN-β increases anti-inflammatory Th2 cytokines in PR and causes a greater shift from Th1 to Th2 in PR than in CR. During exacerbations, IFN-β injection induced a mix of Th1 and Th2 cytokines.

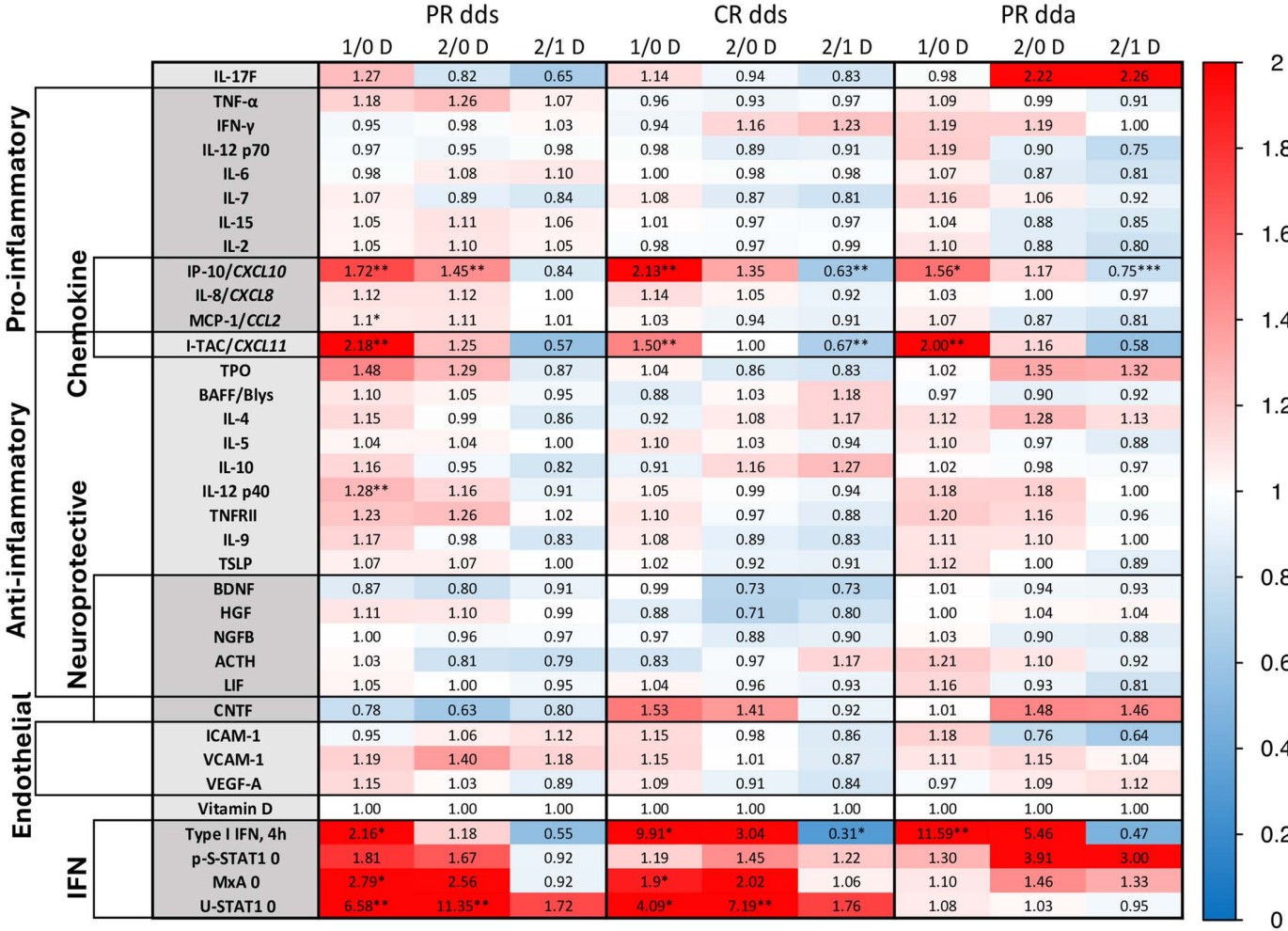

**Multiplex analysis of IFN effects in MS - Lei & Olcer et al. PLOSone 2025**

**Fig 9. Serum protein levels after double-dose IFN-β injections in RRMS.**

### 3. Endothelial (EC) and adhesion markers: Elevated by IFN-β injection during exacerbations

In all stable patients, double-dose and single-dose IFN-β injections tended to induce serum endothelial and adhesion proteins at 1 and 2 days (Fig 9, S7 Fig). During exacerbations, double-dose IFN-β tended to elevate serum ICAM1 and VCAM-1 at 1 day and VCAM at 2 days. This parallels IFN-β potentiation of TNF-induced VCAM release from EC *in vitro* [30]. Single-dose IFN had minimal effects (S7 Fig).

ICAM-1 on inflamed EC is bound by VLA-4 (expressed on activated and memory T cells) and by LFA-1 (on macrophages and B and T cells). Their serum levels increased after IFN-β injection during exacerbations, suggesting a protective mechanism of 16 MU IFN-β during disease activity. Soluble ICAM-1 inhibits T cell activation, and sVCAM-1 blocks VLA-4-mediated adhesion of lymphocytes to BBB EC. Activated endothelial cells also shed the ectodomains of ICAM-1 and VCAM-1. Clinically, high serum levels of shed adhesion proteins correlate with fewer Gd + MRI lesions during IFN-β therapy [30].

### 4. Neurotrophic proteins: Elevated by IFN-β injection during exacerbations

Double-dose and single-dose IFN-β in all stable MS, PR, and CR did not significantly induce serum neurotrophic factors (Fig 9; DNS). During exacerbations, however, 16 MU injection tended to induce ACTH at 1 and 2 days (p = 0.06 4 digits). Single-dose injections tended to elevate neurotrophic factors at 1 day in stable PR compared to CR (S7 Fig). Double-dose IFN was more potent than single-dose IFN-β, and induced relatively more BDNF, CNTF, and HGF in all patients at 1 day, and more HGF in PR at 2 days (DNS).

Serum neurotrophic proteins arise from multiple sources. Circulating PBMC produce small amounts of ACTH, BDNF, CNTF, HGF, and NGF, but most neurotrophic proteins in serum are derived from the pituitary, peripheral nerves, CNS, and stromal cells [31,32].

### F. Tolerability of IFN-β injections

Patients habitually injected IFN-β at night but changed to 8 AM injections in clinic for phlebotomy. 16 of 69 (23%) reported tolerable fatigue, myalgias, warmth, or chills following AM injection. In PR, side effects increased with higher doses and disease activity: single-dose IFN-β in stable MS (15%), double-dose in stable MS (23%), and double-dose in active MS (38%) (3-group $\chi^2$ = 4.173, p = 0.0411), similar to [25,33]). Serum type I IFN activity before injection, and serum IFN and cytokines after injections, did not correlate with side effects.

### Neutralizing antibodies to IFN-β (NAb)

Four of 27 long-term IFN-treated patients were NAb + . In three of 15 PR, serum NAb titers averaged 456 neutralizing units (NU) while clinically stable and averaged 852 NU during exacerbations. In one of 12 CR, the titer was 633 NU. High NAb titers did not affect p-S-STAT1 activation *in vivo* and *in vitro* but tended to prevent MxA induction after single-dose but not double-dose IFN-β injections, *in vivo* and *in vitro*. NAb did not inhibit IFN-β induction of most serum proteins, except for IFN-γ, IP-10, IL-4, and CNTF. In this limited number of NAb+ patients, most MS-relevant cytokines still showed a response to IFN-β.

### II. Long-term IFN-β therapy elevates serum immune and neurotrophic proteins

### A. Therapy-naïve stable and active RRMS *vs*. healthy controls

Clinically stable untreated RRMS patients had decreased pro-inflammatory and anti-inflammatory serum cytokines compared to healthy controls (Fig 10, S8 Fig). Neurotrophic factors had mixed changes. Exacerbating untreated patients, in contrast, had levels of pro-inflammatory cytokines that were similar to levels in healthy controls, but most anti-inflammatory proteins remained low. Overall, untreated exacerbating patients had a higher pro-inflammatory profile than stable patients (Fig 10, right column).

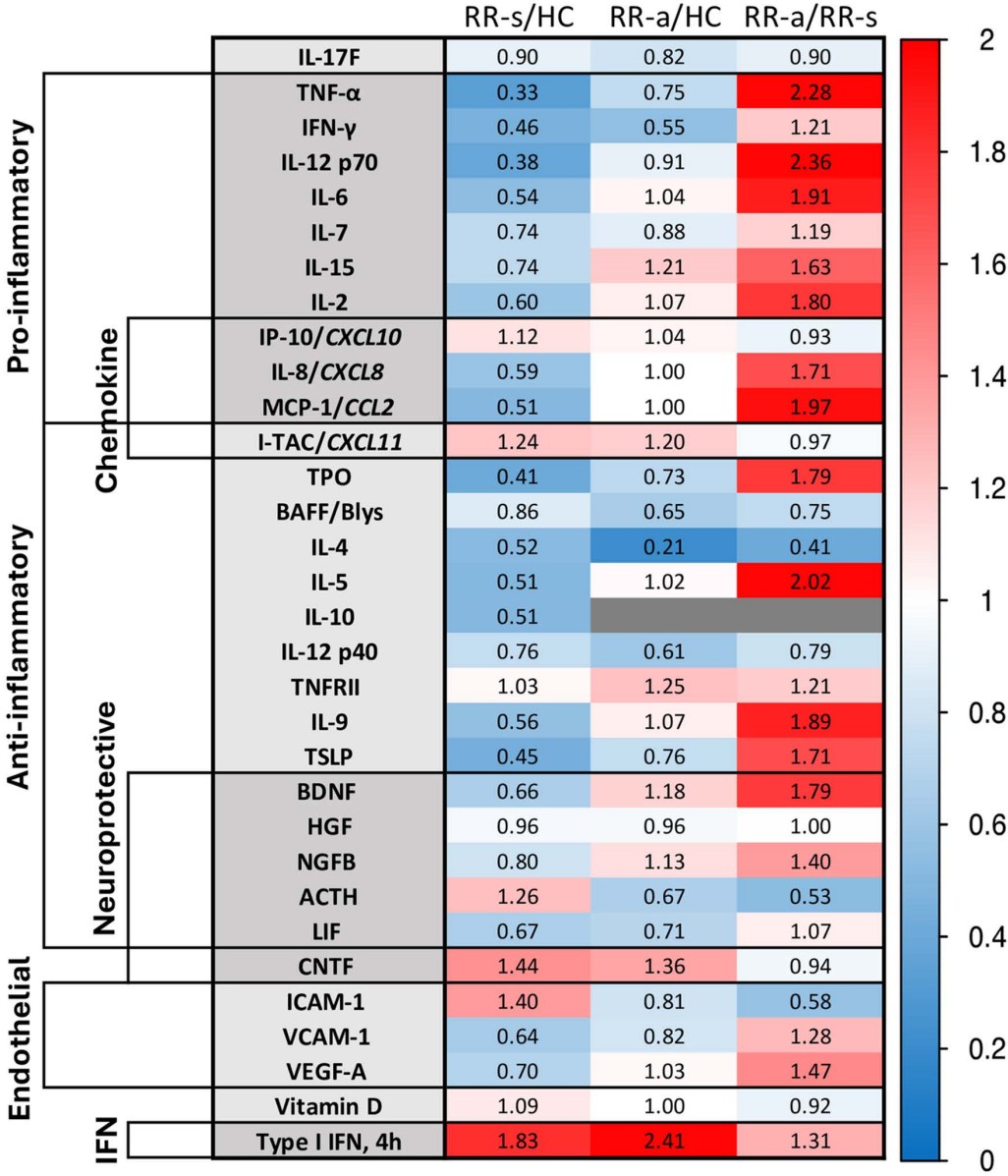

**Fig 10. Serum proteins in therapy-naïve MS *vs*. healthy controls.**

It is striking that levels of almost all proteins were lower in untreated MS compared to HC, even during exacerbations. This argues that Hans Link's concept of a "cytokine storm" during exacerbations [34] is relative only to subnormal levels in stable MS. Exacerbating untreated patients here, compared to stable untreated RRMS, exhibited higher levels of Th1 cytokines, trends for higher Th2 cytokines, and higher homeostatic regulators (Fig 5). Some neurotropic factors were elevated during exacerbations, illustrating the yin/yang nature of MS inflammation and repair. Disruption of immune balance is described in Results, Section III, below.

B.Long-term IFN-β-treated RRMS after therapy washout show elevated cytokines *vs*. therapy-naïve RRMS and HC

Long-term IFN-β therapy elevated serum levels of many proteins to a new baseline. Clinically stable IFN-β-treated PR and CR after washout, compared to healthy controls, tended to have higher levels of pro-inflammatory chemokines and Th1 cytokines, but also expressed multiple Th2 cytokines (Fig 11; S9 Fig). *In vitro* induction of type I IFN was excessive in PR and low in CR compared to HC.

Stable IFN-β-treated PR and CR patients, compared to stable therapy-naïve RRMS, had much higher levels of Th1 chemokines and cytokines but also more Th2 chemokines and regulatory cytokines, homeostatic immune regulators, endothelial markers, and neurotrophic factors (Fig 11). Post-washout levels were consistently replicated between dds and sds (DNS).

Exacerbating IFN-β-treated MS, compared to HC and exacerbating therapy-naive patients, had higher levels of a limited number of chemokines and Th1 cytokines, as well as several elevated Th2/regulatory cytokines (Fig 11). Overall, MS exacerbations significantly reduced IFN-β induction of cytokines and neurotrophic factors.

Thus, long-term IFN-β therapy elevated multiple cytokines, in line with its antiviral and immunoregulatory functions. We next studied the individual and class-grouped regulatory balance of these proteins.

| | | PR-dds | | CR-dds | | PR-dda | |
|---|---|---|---|---|---|---|---|
| | | PR-s / HC | PR-s / Ø-s | CR-s / HC | CR-s / Ø-s | PR-a / HC | PR-a / Ø-a |
| Pro-inflammatory | Chemokine | IL-17F | | | | | |
| | | IL-17F | 3.42 | 3.78 | 1.57 | 1.74 | 1.53 | 1.87 |
| | | TNF-α | 1.00 | 3.04* | 1.84* | 5.56** | 0.85 | 1.13 |
| | | IFN-γ | 3.30 | 7.22 | 3.25 | 7.10 | 2.00 | 3.61 |
| | | IL-12 p70 | 0.81 | 2.11 | 0.76 | 1.97 | 0.53 | 0.59 |
| | | IL-6 | 1.41 | 2.59 | 1.32 | 2.42 | 0.93 | 0.89 |
| | | IL-7 | 0.97 | 1.31 | 0.94 | 1.27 | 0.79 | 0.90 |
| | | IL-15 | 0.99 | 1.33 | 0.94 | 1.27 | 0.75 | 0.62 |
| | | IL-2 | 1.94 | 3.25** | 2.31* | 3.88** | 1.76 | 1.64 |
| | | IP-10/CXCL10 | 1.44 | 1.29 | 1.70* | 1.52* | 1.66 | 1.60* |
| | | IL-8/CXCL8 | 0.88 | 1.5~ | 0.94 | 1.60 | 0.95 | 0.94 |
| | | MCP-1/CCL2 | 0.86 | 1.68 | 0.86 | 1.69 | 0.80 | 0.80 |
| Anti-inflammatory | Neuroprotective | I-TAC/CXCL11 | 2.22 | 1.79** | 3.29 | 2.66 | 2.30 | 1.92 |
| | | TPO | 1.24 | 3.04 | 0.96 | 2.36 | 0.65 | 0.89 |
| | | BAFF/Blys | 1.28 | 1.49 | 0.93 | 1.08 | 1.18 | 1.83 |
| | | IL-4 | 2.06 | 3.93 | 2.38 | 4.54* | 1.77 | 8.25 |
| | | IL-5 | 0.91 | 1.79 | 1.08 | 2.12 | 0.63 | 0.62 |
| | | IL-10 | 1.01 | 1.98 | 1.14 | 2.25 | 0.82 | |
| | | IL-12 p40 | 2.27 | 2.97*** | 1.84 | 2.41* | 2.19 | 3.62*** |
| | | TNFRII | 1.41 | 1.37 | 2.18* | 2.11* | 1.47 | 1.18 |
| | | IL-9 | 2.00 | 3.54*** | 1.44 | 2.56** | 2.00 | 1.88 |
| | | TSLP | 0.90 | 2.02 | 1.05 | 2.35~ | 0.70 | 0.91 |
| | | BDNF | 2 | 2.56 | 1 | 2.16~ | 1.42 | 1.21 |
| | | HGF | 1.28 | 1.34 | 1.37 | 1.43 | 1.09 | 1.14 |
| | | NGFB | 1.00 | 1.24 | 0.61 | 0.76 | 0.81 | 0.72 |
| | | ACTH | 1.07 | 0.85 | 1.08 | 0.86 | 0.76 | 1.13 |
| | | LIF | 1.32 | 1.98 | 1.11 | 1.67 | 1.01 | 1.41 |
| Endothelial | | CNTF | 1.2 | 0.81 | 1.0 | 0.66 | 0.45 | 0.33 |
| | | ICAM-1 | 1 | 0.48 | 1 | 0.93 | 1 | 1.27 |
| | | VCAM-1 | 1 | 1.69 | 2 | 2.68~ | 1.37 | 1.67 |
| | | VEGF-A | 1.05 | 1.50 | 1.14 | 1.63 | 0.72 | 0.70 |
| IFN | | Vitamin D | 1.01 | 0.92 | 0.96 | 0.88 | 0.96 | 0.96 |
| | | Type I IFN, 4h | 2.33 | 1.27** | 0.28 | 0.15*** | 1.21 | 0.50 |

**Fig 11. Serum proteins in double-dose IFN-β-treated RRMS: Long-term effects.**

III. **Positive correlation of pro-inflammatory and immunoregulatory serum protein levels in HC and in MS after short-term and long-term exposure to IFN-β, but poor correlation in untreated MS**

A. **Serum protein correlations are strong in HC, but weak in untreated MS**

In healthy controls, levels of almost all Th1 and Th2 cytokines and neurotrophic proteins were positively correlated (Fig 12A, subclass averages in upper right of diagonal), presumably reflecting a state of balanced immune control, *e.g.*, all high or all low in an individual. Serum vitamin D was negatively correlated with almost all targets. This suggests that high serum vitamin D is linked to low levels of most cytokines in HC.

Untreated stable MS patients, in contrast, had less correlated CXCL10, IL-9 and EC markers (Fig 12B; more blue *left*, lower averages *right*). Vitamin D levels correlated with most targets, perhaps linked to its induction of Th2 regulatory immunity [7]. Active MS patients exhibited weaker or negative correlations among Th1 and Th2 cytokines, neurotrophic factors, and endothelial proteins (Fig 12C). Within cytokine clusters, Th1 cytokines were largely intercorrelated; Th2 cytokines were less so. Immune dysregulation in MS reflects more than Th1 and T2 cytokine imbalance; it is a multi-body interaction even within Immune subclusters, in keeping with an immune regulatory defect in MS [4,35].

B. **IFN-β treatment promotes correlation of serum protein levels in MS**

Long-term exposure to IFN-β led to balanced serum Th1 cytokine levels. In stable PR after washout, serum Th1 chemokines and cytokines correlated with each other and with some Th2 cytokines (prominently IL-5, TPO, and TSLP), neuroprotective proteins (ACTH, LIF, and NGF), type I IFN, and vitamin D (Fig 13A). In CR after washout, Th1 cytokines correlated with each other, with Th2 cytokines, and with neuroprotective proteins, but now inversely with vitamin D (Fig 13B). Th2 cytokines were more intercorrelated in stable CR than in stable PR. Post-washout serum type I IFN and signaling molecules were not correlated in CR, likely due to the very low IFN signature in CR. In exacerbating PR after IFN washout, correlations were similar to stable PR for Th1, Th2, neuroprotective proteins, and vitamin D (Fig 13C).

One and two days after IFN-β injections, post-washout patterns were maintained in all groups (S10A–C Figs and DNS), but with weaker correlations in active PR among IFN signaling proteins, lower right), paralleling reduction of gene expression in IFN signaling pathways during exacerbations [3]. This profile suggests that long-term IFN-β therapy forged a new steady state that persisted during the short-term IFN-β effect. Even though IFN-β therapy induces higher levels of many cytokines compared to untreated MS, they are in better balance. This resembles the high correlation in healthy controls, and contrasts with the highly dysregulated cytokine environment in untreated MS (Fig 12A). Imbalanced serum protein correlations in untreated MS and improvement with IFN-β therapy parallel the extensive dysregulation of mRNA expression in PBMC from untreated MS and the return to near-normal expression with long-term IFN-β therapy [4,47].

C. **Serum vitamin D *vs*. serum type I IFN levels and IFN-β-induced signaling proteins**

Vitamin D enhances type I IFN signaling and causes a shift from Th1 to Th2 immunity *in vitro* [7]. We investigated the effect of serum vitamin D on serum IFN levels and responses to IFN injections.

**Serum vitamin D correlates with serum type I IFN in PR, post-washout**

Serum levels of 25-hydroxy vitamin D in IFN-β-treated MS patients averaged 28.8±2.7 ng/ml (72.0±6.8 nmol/L; median, 29 ng/ml), near the 30 ng/ml "sufficient" threshold. Stable PR (30.7±3.4 ng/ml) did not differ from stable CR (27.8±2.4 ng/ml). Serum vitamin D was equivalent between healthy controls (29.0±3.6 ng/ml), untreated stable RRMS (32.9±3.6 ng/ml), and exacerbating MS (30.2±4.0 ng/ml). Vitamin D levels were equivalent in MS and HC, reflecting current widespread oral vitamin D supplementation. Ten years before this study, vitamin D levels were lower in MS [2].

Serum vitamin D levels correlated positively with post-washout type I IFN activity in dds PR (r=0.63, p<0.02), and less so in CR, where IFN was often undetectable, r=0.23, NS), but not in exacerbating PR (r=0.04, NS) (Fig 13A–C, single

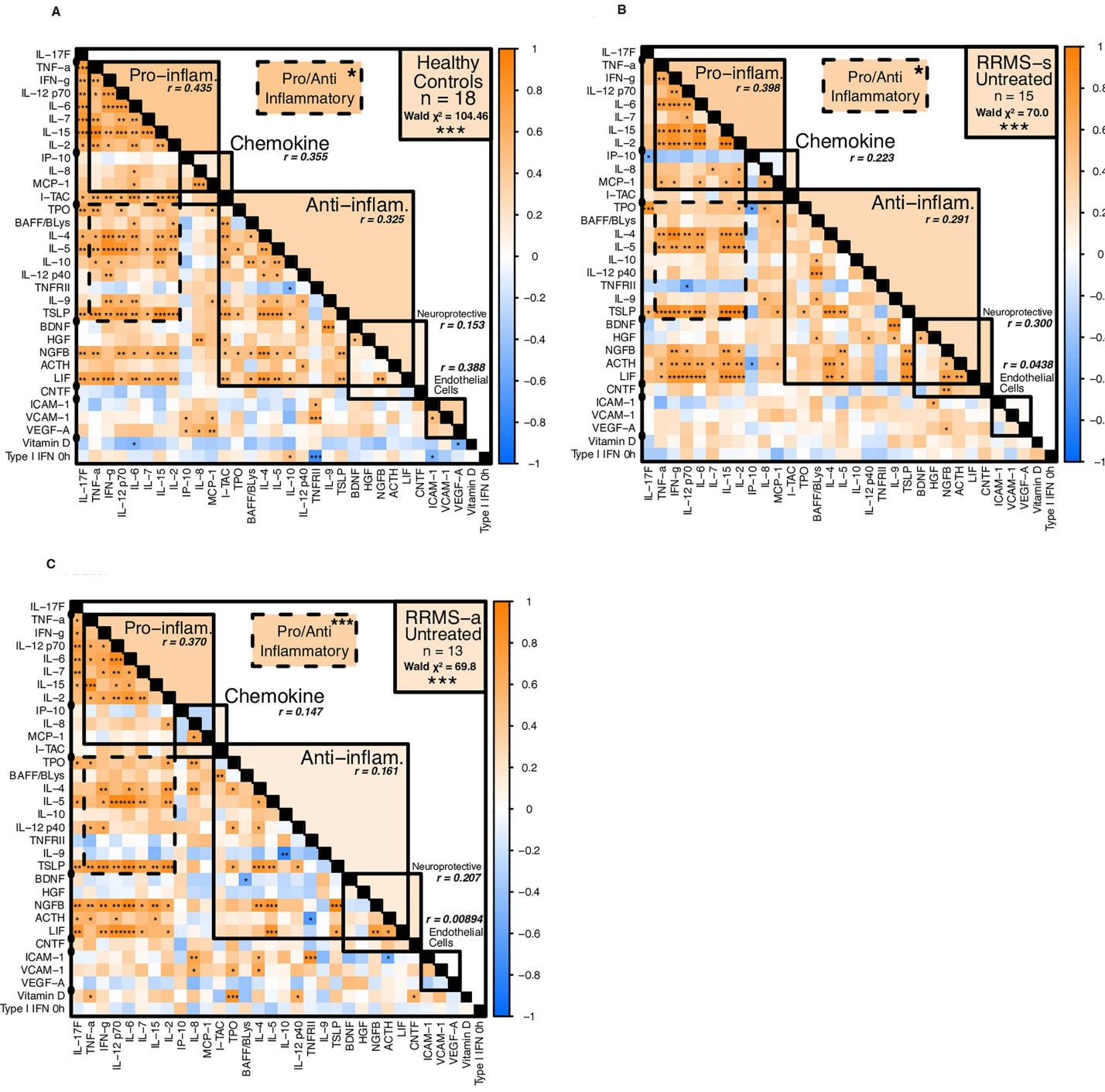

**Fig 12. Correlation matrices of serum protein levels in HC and untreated RRMS.**

pixel). Intracellular MxA levels, reflecting long-term IFN induction, were positively correlated in all groups. In post-washout stable PR, serum vitamin D levels correlated positively with some pro-inflammatory, anti-inflammatory, and neurotrophic proteins. It is possible that higher IFN responses in PR shift immunity to a Th1 response, causing the mixed Th1/Th2

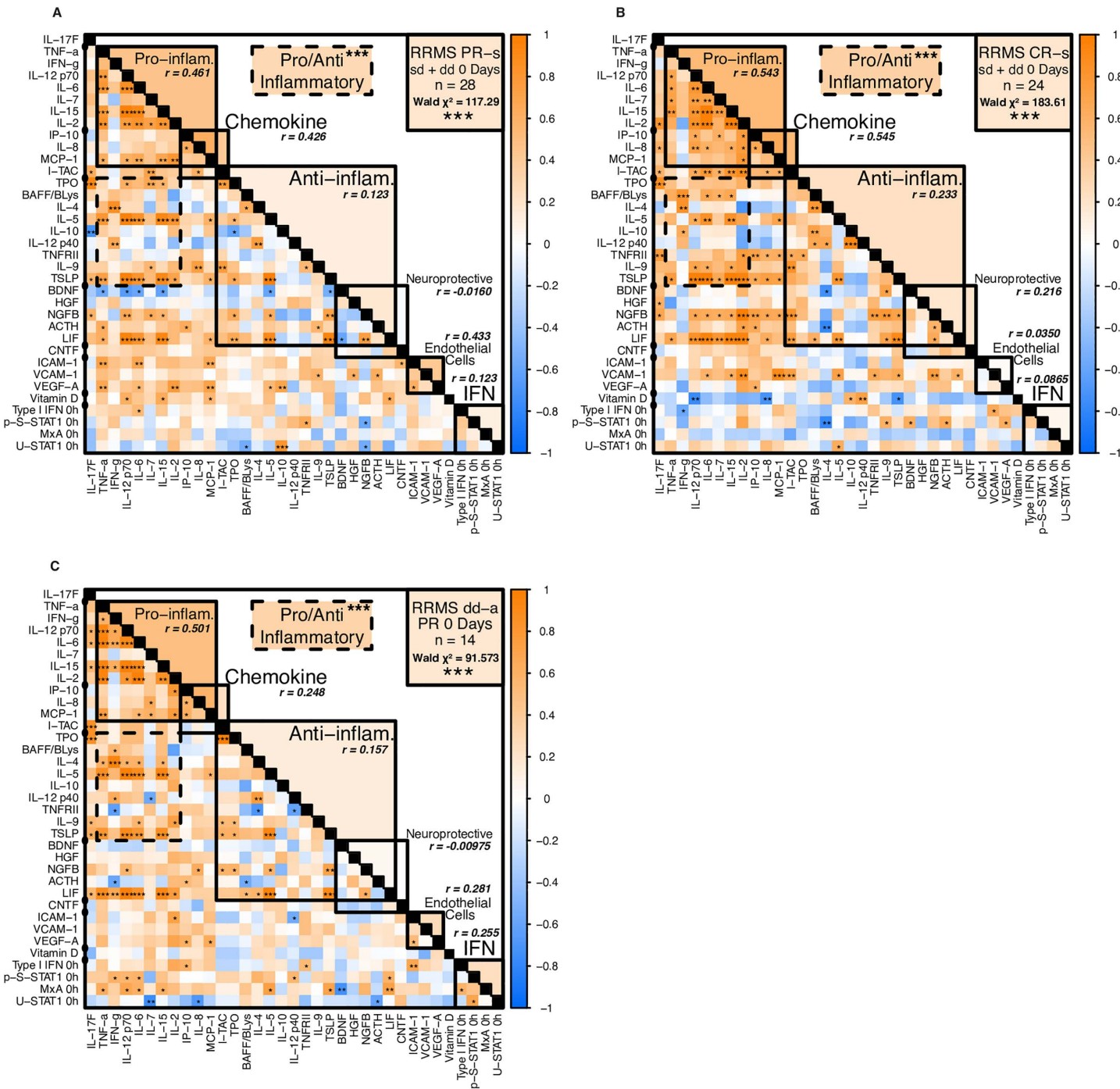

**Fig 13. Correlation matrices of serum protein levels after washout in IFN-β-treated PR and CR.**

profile. In stable CR, however, vitamin D was negatively correlated with inflammatory cytokines and positively with two anti-inflammatory proteins, IL-10 and IL-12p40. After IFN therapy washout, vitamin D was linked to increased serum neurotrophic proteins in PR and to a shift from Th1 to Th2 immunity in CR more than in PR.

## Serum vitamin D correlates with serum cytokines in PR but not in CR, after IFN-β injections

After double-dose IFN-β injections in stable PR, higher serum vitamin D correlated with higher Th1 chemokines and cytokines and neurotrophic factors at 1 day (S9A Fig). During IFN-β-treated exacerbations, vitamin D correlations were weak for all targets (S9C Fig). In CR, high vitamin D correlated with lower Th1 and Th17, but had mixed effects on Th2 cytokine levels at 1 and 2 days (S9B Fig and DNS).

After single-dose IFN-β injections in stable PR, high vitamin D again tended to correlate with increases in many inflammatory and noninflammatory cytokines and LIF and NGF (DNS). In CR, however, high serum vitamin D correlated with lower Th1, and now with more Th2 cytokines at day 1 (IL-4, IL-10, IL-12p40, DNS). This suggests the standard IFN-β dose has potent immunoregulatory effects in CR.

MS progression and brain repair during IFN therapy could be influenced by vitamin D-enhanced induction of neurotropic proteins, even in PR. In CR receiving standard-dose and double-dose IFN-β, vitamin D correlated with lower Th1, suggesting an anti-inflammatory clinical benefit of high serum vitamin D in this group.

## Discussion

In untreated MS, serum cytokines are at unexpectedly low levels and are also poorly correlated with each other. During IFN-β therapy, serum type I IFN and IFN-induced PBMC proteins were linked to future clinical response, *i.e.*, levels were paradoxically higher in stable PR than in CR. IFN-β injections activated IFN signaling and had a long-term priming effect to enhance subsequent IFN responses. With IFN-β therapy, cytokine levels and their balance increased.

### IFN-β injections activate STAT1, with antiviral and immunoregulatory effects

Our data suggest mechanisms for immune and epidemiologic aspects of MS. Untreated MS patients have low serum type I IFN levels, low levels of activated p-S-STAT1 transcription factor, and low ISG mRNA compared to HC [3,5,6,10]. These abnormalities are more extreme during exacerbations and progression and may affect antiviral and immune regulation in untreated MS. In support, long-term injection of IFN-β elevates levels of p-S-STAT1, MxA, U-STAT1, and type I IFN. The change in IFN responses during IFN-β therapy correlates with correction of immune regulatory defects in MS [2,4,36].

An inflammatory residue from virus infections in MS is suggested by high anti-measles, rubella, herpes zoster, and EBV titers in serum and CSF [1,2] and by excessive T cell responses to measles in the MS-affected twin [37]. Despite low p-S-STAT1 levels, virus infections in MS are half as frequent as in healthy controls and are most reduced in patients with high EDSS scores, based on studies predating widespread use of MS disease-modifying therapies [38–40].

The paradox of low p-S-STAT1 and poor immune regulation, yet better antivirus immunity, may arise from compensatory elevation of p-Y-STAT1 and induction of antiviral and pro-inflammatory genes. When p-S-STAT1 formation in murine bone-marrow-derived macrophages is eliminated by an S727A mutation, IFN-α/β and IFN-γ still induce a large subset of p-Y-STAT1-activated genes that enhance NK cell killing of tumor cells [41] and antiviral responses [42]. p-Y-STAT1 formation is robust and tunable [43]. Four hours after IFN-β injection, we see induction of p-Y-STAT1-responsive genes (*e.g.*, *GBP*, 10-fold increase; *TAP1*, 3-fold; *IRF1*, 2-fold; genes identified in [42]). Expression of these genes is excessive in resting PBMC from untreated MS (gene accession from [4]). Expression of these genes in MS PBMC is downregulated by long-term IFN-β therapy (*GBP*, 5-fold decrease; *TAP1*, 2.5-fold), and the new expression set point is close to that of healthy controls. However, not all anti-viral genes fit this categorization, as low MxA levels parallel low p-S-STAT1 in untreated MS [3].

Type I IFNs elevate unphosphorylated STAT1 [44], providing more substrate for p-STAT1 formation. The amount of available U-STAT1 was higher in PR than in CR (Fig 6). U-STAT1 itself induces a subset of IFN-regulated genes that target viruses and that protect against chemotherapy- and radiation-induced DNA damage [44,45]. Here, IFN injections and restimulation in vitro induced U-STAT1 in both PR and CR, but more in stable PR, potentially enhancing IFN-signaling, anti-viral, and neuroprotective effects.

## High serum type I IFN levels are linked to future MS exacerbations and response to IFN therapy in MS and NMO

Serum type I IFN levels are low in untreated MS [5]. IFN injections had clinical benefit in both CR and PR during long-term IFN-β therapy. We now find that serum type I IFN was higher in PR than in CR after therapy washout (Fig 8). Thus, serum type I IFN levels are linked to response to IFN therapy. IFN-β injection also "primes" the subnormal type I IFN system in MS. This allows a second dose of IFN-β to facilitate faster, higher, and stronger gene induction (*citius, altius, fortius*) [44,46–48].

There is a spectrum of IFN responses in demyelinating disease. CR patients, who had lower serum type I IFN levels, had no attacks over at least 5 years. PR patients, who had higher baseline type I IFN levels, had only partial clinical responses to IFN-β therapy, but fewer attacks than expected without treatment. Paralleling our data, monocytes from MS clinical non-responders to IFN-β therapy show high pre-treatment ISG expression [49]. A more extreme example of CNS inflammation with high serum IFN levels is neuromyelitis optica (NMO), where antibodies to the aquaporin-4 water channels of astrocyte foot processes cause severe inflammation, neuron destruction, and secondary demyelination. In NMO and lupus (SLE), serum type I IFN levels are 8-fold above normal [SLE, average 3.10 IFN units; HC, 0.40 units], and IFN-β injections cause exacerbations [5]. In MS, in contrast, serum IFN levels are 4-fold lower than healthy controls [MS 0.10 units] and IFN-β injections ameliorate the disease. In CR, long-term IFN-β injections elevated serum type I IFNs ~3-fold from a very low endogenous washout level (0.29 ± 0.12 IFN units). In PR on long-term therapy, washout levels were higher (5.2 ± 2.4) and similar to levels in untreated SLE [5].

## Chemokines, classic IFN-induced proteins

Chemokines were dose-dependently induced by IFN-β. Imbalanced levels in untreated MS became balanced with therapy, with potential clinical benefit. Chemokines attract pro-inflammatory but also immune-regulatory cells; some chemokines have both functions.

Pro-inflammatory CXCL10/IP-10 and CXCL8/IL-8 enhance Th1 cell migration to sites of inflammation. CXCL10 is strongly induced by IFN-β and is toxic to oligodendrocytes [50]. CXCL8 attracts Th1 cells but also enhances post-inflammatory repair and induces VEGF-A, which promotes angiogenesis and bone growth. CXCL8 receptors are present on oligodendrocytes and CXCL8 induces differentiation of oligodendrocyte progenitor cells and myelin formation [51], of potential benefit in brain repair during CNS inflammation. CXCL8 is released from endothelial cells (EC) and PBMC. EC shedding of CXCL8 *in vivo* may explain the high level seen with IFN therapy, differing from *in vitro* experiments, where IFN-β inhibits CXCL8 secretion by PBMC [4,52]. CCL2/MCP-1 has many pro-inflammatory effects (Methods). However, CCL2 and CXCL11/I-TAC 1 [53] enhance Th2 cell migration and CXCL11 may inhibit EAE [54].

## Cytokines and neurotrophic proteins are dose-dependently induced by IFN-β injection—implications for therapy in MS and NMO

Th2 and regulatory cytokines predominated after double-dose IFN-β injection. IL-12 p40, a monocyte product with Th2/anti-inflammatory role in MS, was induced in PR, and IL-12p70 (Th1 role) fell in CR. Levels of these two cytokines were negatively correlated (r = −0.384, p = 0.024). Supporting these data, IFN-β-1a induced serum IL-12p40 in a different group of stable RRMS patients, measured with a different multiplex array [47]. The IL-12p40 homodimer blocks effects of pro-inflammatory IL-12p70 and IL-23 [17], blocks induction of Th1 and Th17 cytokines [55], and increases CD4 Treg function [56]. Clinically, anti-IL-12p40 monoclonal antibodies (MAb) worsen MS [57]. We suggest that anti-IL-12 MAb induces exacerbations by blocking anti-inflammatory p40 homodimers.

TNFRII was induced at 24 hours by 16 MU IFN-β in all patients, paralleling IFN induction of its gene, *TNFRSF1B* [4]. In MS, IFN-β-induced TNFRII levels correlate with fewer T2 MRI lesions [58]. Circulating TNFRII, shed from Treg and EC, inhibits secretion of pro-inflammatory TNF-α and IL-6 [59]. TNFRII, the high-affinity TNF-α receptor, is abundantly

expressed on CD8 Treg and CD4 Treg, enhancing their activation by membrane-bound TNF-α [60,61]. In MS, CD8 and CD4 regulatory functions are compromised [1,2,62], and high serum TNFRII levels may enhance Treg function. TNF-RII also activates CNS astrocytes to secrete LIF, which induces neuronal survival and oligodendrocyte maturation and regeneration [63]. TNF blockers affect both TNFRI and TNFRII; effects are likely to vary with dose of anti-TNF, variation in immune cell subsets, and MS disease activity. Clinically, MAb to TNF-α downregulate inflammation in connective tissue diseases [60]. However, anti-TNF-α antibodies worsen MS [2,60], likely by preventing activation of TNFRII on already-compromised MS regulatory T cells.

Clinical response to IFN-β depends on the form of MS (relapsing/progressive), disease state (stable/active), genetics, and environment (smoking, serum vitamin D level, BMI, exercise, and past EBV infection). With standard injections of type I IFN, benefit evolves with the disease -- best in early MS, lower in moderate-EDSS MS [64], and still lower during exacerbations and in high-EDSS and progressive MS [3,5,48,64]. Double-dose IFN-β injection induced some serum proteins more than single-dose injections. During exacerbations, double-dose IFN-β induced pro-inflammatory IL-2, IL-6, and IL-7, but also increased potentially beneficial type I IFN and Th2 cytokines (IL-10, TNFRII), soluble adhesion molecules (ICAM, VCAM), and neurotropic (HGF, ACTH) proteins (Fig 9, S7 Fig). High-dose IFN-β does not have additional clinical benefit in reducing attacks in RRMS, although repair on MRI is enhanced [33,65]. Pegylated IFN-β-1a, with longer half-life than native IFN-β-1a, also induces Th2 and neurotrophic proteins [47]. The present study suggests that different doses of IFN-β could overcome IFN resistance during exacerbations

The protein assays herein complement assays of gene expression [4,7,14,4 [9,47,48] and alternative RNA splicing [66] that were replicated in separate cohorts of RRMS patients. Gene expression, alternative splicing, and protein expression are all dysregulated and imbalanced in untreated MS, and all are corrected by IFN-β therapy. Although IFN-β is not the most potent MS therapy on average, it does correct MS-linked dysregulation in the IFN signaling system.

## Serum Vitamin D levels may enhance immune regulation during IFN-β therapy

Vitamin D enhances activation of p-Y-STAT1 to potentiate IFN-β signaling and antiviral responses [7]. COVID-19, a virus that severely disrupts IFN signaling, is milder in patients with adequate vitamin D levels [67] and in MS patients treated with IFN-β before onset of COVID infection [68,69]. Vitamin D also potently elevates immune-regulatory Th2 proteins and is linked to fewer MS exacerbations during IFN-β therapy [4,7,69]. Here, post-washout serum vitamin D levels correlated positively with serum type I IFN levels in stable PR (Fig 13A). After double-dose IFN-β injections in stable CR, serum vitamin D correlated with decreased Th1 cytokines and with increased Th2 cytokines (Fig 13B), suggesting an additive immunoregulatory benefit of IFN-β and vitamin D in stable MS. During exacerbations, vitamin D correlated with IFN-β-enhanced levels of Th1 cytokines, but also with induction of neurotrophic proteins, potentially enhancing CNS repair.

Low serum vitamin D levels augur a greater chance of developing MS and are linked to more exacerbations and faster progression in untreated MS [70–71]. Vitamin D supplementation, especially in patients with low baseline serum vitamin D levels, prevents a second demyelinating attack [72,73]. During IFN-β-1b therapy, subnormal serum levels of vitamin D correlate with more exacerbations, progression, MRI lesions, and brain atrophy [74,75]. In support, vitamin D is additive with IFN-β-1b therapy in reducing MRI lesions [74]. A 20 mg/ml serum increment of 25-hydroxy vitamin D reduces new active MRI lesions, and loss in brain volume by 0.41%, relapse rate by 57%, and progression by 17% [75], perhaps because vitamin D enhances IFN signaling and induces a Th2 shift [7]. In contrast to IFN-β, glatiramer acetate causes a strong shift to Th2 immunity so supplemental vitamin D does not enhance its benefit in patients with adequate baseline serum vitamin D levels of 29 ng/ml [76].

There has been a recent increase in serum vitamin D in MS patients *vs.* historically low levels [2,77]. Clinical trials of IFN-β and other MS therapies in the 1990's may not reflect the current immune ecology in MS, where higher levels of serum vitamin D now may enhance responses to IFN-β therapy and vaccinations [7,72–75,77].

## Long-term IFN-β therapy elevates and balances serum proteins in multiple biological pathways and primes responses to a subsequent IFN injection

Untreated MS patients have surprisingly low and dysregulated serum cytokine levels compared to HC, but levels rise during exacerbations. Individual protein levels within and between immune response groups (*e.g.*, Th1, Th2) were often in opposite directions. Patients after IFN-β therapy washout have higher levels of type I IFNs and Th1 chemokines and cytokines, but also higher Th2 cytokines, homeostatic cytokines, and endothelial cell proteins. Long-term IFN-β therapy moved many of these cytokines and proteins to a more balanced profile than that of untreated MS. The protein changes parallel and extend data showing highly dysregulated RNA expression of 8,000 genes in untreated MS [4] and assays in other MS cohorts [34,44], and other large scale protein arrays [47,48]. This protein dysregulation is beyond a simple Th1/Th2 framework, and indicates there is broad and complex disruption of immunity in untreated MS.

The increased expression of multiple curated, MS-relevant Th1, Th17, Th2, and neuroprotective proteins with IFN-β therapy herein differs from many early reports. These studies measured cytokine production by *in vitro* activation of PBMC or small numbers of serum cytokines and often showed a decrease in Th1 and increase in Th2 cytokines with therapy [1,2]. IFN-β therapy reduction of relapses, progression, cognitive loss, MRI lesions, and prolonged longevity in MS [8], may be linked to stress-induced repair mechanisms [78], and enduring changes in immune and neurotrophic homeostasis.

Long-term IFN-β therapy enhanced responses to a subsequent IFN injection for p-S-STAT1, MxA, U-STAT, and type I IFN itself. During exacerbations, this "priming" was reduced and parallels induction of multiple IFN-induced genes [3,48]. In contrast, faster, higher, and stronger signaling during long-term IFN-β therapy could enhance therapeutic effect, just as it does for antivirus and vaccine responses [46–48,69,77].

## Serum protein levels do not always match RNA expression, and other limitations

IFN-β therapy induced a rise in serum proteins. After IFN-β injection, highly-induced serum chemokine levels correlate with PBMC RNA expression, but many other proteins do not correlate [4]. This RNA/protein dissonance arises from many confounders--serum proteins from non-immune cells and target cells absorption of proteins. Additionally, lack of correlation may arise from post-injection timing of RNA and protein assays, regulation by non-translated mRNA and lncRNA, and large amounts of alternative splicing in untreated MS [66], possibly creating alternatively-spliced protein isoforms not detected by assay antibodies. Immune cell subpopulations also differ in post-transcriptional regulation. For instance, IFN-β shortens the half-life of IL-10 mRNA in monocytes, but prolongs the half-life in T cells [79], perhaps because Argonaut-microRNA complex binding to the 3' UTR changes cytokine half-life [80]. For clinical correlations, detectable MS-linked serum or CSF proteins may be more relevant targets for biomarker panels than short-lived RNA.

Other limitations: 1) A moderate number of patients in each group could reduce statistical significance. We compensate with careful clinical categorization and paired longitudinal design, and compare data to publications describing comparable cohorts. Multiple internal validations indicate the results are not spurious. 2) MRI evaluations and confirmatory MRI for exacerbations were not performed because new MRI lesions are markedly reduced during disease-modifying therapy, requiring very large numbers of patients. Moreover, correlation between Gd+ lesions and clinical activity in the best case, untreated MS, is only 0.2 [1,2]. 3) The immediate transition from untreated to treated MS was not captured, because the first titrated dose of IFN-β would be only ¼ dose. 4) Changes over time during clinical stability were not measured. 5) Combinatorial effects of vitamin D and symptomatic MS therapies in responders and partial responders to IFN-β and other disease-modifying therapies need rigorous evaluation.

## Conclusions

Injection of IFN-β corrects subnormal levels of serum type I IFN and induces Th2 cytokines and neurotrophic factors. Using IFN injections to probe the abnormal IFN signaling seen in MS, we find that the IFN profile reflects the clinical state of MS and predicts future disease activity in stable MS. The lowest baseline serum type I IFN levels were in clinical complete responders to IFN therapy. IFN-β injections prime responses to subsequent type I IFN exposure, enhancing antiviral

response and likely immune control. With long-term IFN-β therapy, the dysregulated type I IFN system and cytokine balance changes towards a normal profile, appearing to balance dysregulated immunity. This correction is not perfect, and IFN-β and all other current MS therapies are only partially effective. Understanding inflammatory mechanisms and cytokine balance can suggest combination therapies. For instance, agents that prompt a shift to Th2 immunity, such as vitamin D [4,7,72,74,75,77] and cAMP inducers [2,79,81,82] may enhance benefit of IFN-β and other MS therapies.

## Supporting information

**S1 Fig. Study design.**
(TIF)

**S2 Fig. 36 protein targets.**
(TIF)

**S1 Table. p-S-STAT1, MxA, and U-STAT1 levels before and after *in vivo* and *in vitro* IFN-β induction.**
(XLSX)

**S2 Table. Absolute values.**
(XLSX)

**S3 Fig. MxA levels 48 hours after *in vitro* IFN-β induction.**
(TIF)

**S4 Fig. MxA levels 24 + 48 hours (AUC) after *in vitro* IFN-β induction.**
(TIF)

**S5 Fig. U-STAT1 levels 48 hours after *in vitro* IFN-β induction.**
(TIF)

**S6 Fig. U-STAT1 levels 24 + 48 hours (AUC) after *in vitro* IFN-β induction.**
(TIF)

**S7 Fig. Serum protein levels after single-dose IFN-β injection in RRMS.**
(TIF)

**S8 Fig. Serum proteins in stable or active therapy-naïve MS *vs*. healthy controls.**
(TIF)

**S9 Fig. Serum proteins after IFN-β therapy washout *vs*. therapy-naïve MS and healthy controls.**
(TIF)

**S10 Fig. Correlation matrices of serum protein levels one day after double-dose IFN-β injections.**
(TIF)

**S11 Fig. Raw images.**
(PDF)

## Acknowledgments

Thanks for IFN activity assays by Tim Niewold, Department of Medicine, University of Chicago; NAb assays by Florian Deisenhammer, Department of Neurology, University of Innsbruck, Innsbruck, Austria; and the University of Chicago Genomics Core facility for multiplex assays. Statistical help from Riyue Bao, University of Chicago Bioinformatics,

Nicholas P Reder, University of Washington and Alpenglow Biosciences, Seattle, WA, and Elena Badillo-Goicoechea, Senior Data Scientist, Biostatistics Laboratory, University of Chicago Department of Public Health Sciences. Standardization of the multiplex assay was performed by Brendan Yee (Panomics/Affymetrix, Santa Clara, CA) to obviate antibody cross-reactivity and to define expression levels. Anti-MxA antibodies were generously provided by Stefan Lanker, Biogen, Boston, MA. Summary figure created with BioRender.com.

## Author contributions

**Conceptualization:** Xuan Feng, Anthony T. Reder.

**Data curation:** Lei Li, Maya Olcer, Zhe Wang, Yaerin Song, Jeffrey Ke, Xuan Feng, Anthony T. Reder.

**Formal analysis:** Lei Li, Maya Olcer, Zhe Wang, Jeffrey Ke, Xuan Feng, Anthony T. Reder.

**Funding acquisition:** Xuan Feng, Anthony T. Reder.

**Investigation:** Lei Li, Yaerin Song, Xuan Feng, Anthony T. Reder.

**Methodology:** Maya Olcer, Xuan Feng, Anthony T. Reder.

**Project administration:** Xuan Feng, Anthony T. Reder.

**Resources:** Xuan Feng, Anthony T. Reder.

**Software:** Maya Olcer, Jeffrey Ke.

**Supervision:** Xuan Feng, Anthony T. Reder.

**Validation:** Lei Li, Maya Olcer, Zhe Wang, Yaerin Song, Xuan Feng, Anthony T. Reder.

**Visualization:** Lei Li, Maya Olcer, Jeffrey Ke, Anthony T. Reder.

**Writing – original draft:** Maya Olcer, Xuan Feng, Anthony T. Reder.

**Writing – review & editing:** Lei Li, Maya Olcer, Zhe Wang, Yaerin Song, Xuan Feng, Anthony T. Reder.

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
