## [Decision Letter · Decision Letter 0]

5 Jan 2025

Dear Dr. Feng,

Thank you for submitting your manuscript to *PLOS One* .  Your manuscript has been reviewed by two experts in the field. I have also briefly reviewed the manuscript, with a focus on the data section. This is interesting and potentially important work. I concur with the reviewers' concerns, and their comments are attached. Please address these concerns thoroughly in your revised version, giving particular attention to those of Reviewer #1. I have noticed that the presentation is somewhat disorganized and cluttered. I suggest including the most important figures and tables in the main manuscript and placing the remaining ones in supplemental materials. Please also revise your abstract to make the sentences more specific. For example, the statement "The balance of Th1, Th2, and neurotrophic proteins was disrupted in therapy-naïve MS" requires clarification regarding the specific nature of the disruption. My decision is a major revision.

We look forward to receiving your revised manuscript.

Kind regards,

Luwen Zhang

Academic Editor

PLOS ONE

2. Please note that your Data Availability Statement is currently missing [the repository name and/or the DOI/accession number of each dataset OR a direct link to access each database]. If your manuscript is accepted for publication, you will be asked to provide these details on a very short timeline. We therefore suggest that you provide this information now, though we will not hold up the peer review process if you are unable.

3. Please ensure that you refer to Figures 7B, 7C, 8B, 8C, 9B, 9C in your text as, if accepted, production will need this reference to link the reader to the figure.

4. Please include a caption for figures 1 and 2.

Additional Editor Comments:

Thank you for submitting your manuscript to PLOS One. Your manuscript has been reviewed by two experts in the field. I have also briefly reviewed the manuscript, with a focus on the data section. This is interesting and potentially important work. I concur with the reviewers' concerns, and their comments are attached. Please address these concerns thoroughly in your revised version, giving particular attention to those of Reviewer #1. I have noticed that the presentation is somewhat disorganized and cluttered. I suggest including the most important figures and tables in the main manuscript and placing the remaining ones in supplemental materials. Please also revise your abstract to make the sentences more specific. For example, the statement "The balance of Th1, Th2, and neurotrophic proteins was disrupted in therapy-naïve MS" requires clarification regarding the specific nature of the disruption. My decision is a major revision.

Reviewers' comments:

Reviewer's Responses to Questions

**Comments to the Author**

1. Is the manuscript technically sound, and do the data support the conclusions?

Reviewer #1: Partly

Reviewer #2: Partly

2. Has the statistical analysis been performed appropriately and rigorously?

Reviewer #1: I Don't Know

Reviewer #2: I Don't Know

3. Have the authors made all data underlying the findings in their manuscript fully available?

Reviewer #1: Yes

Reviewer #2: Yes

4. Is the manuscript presented in an intelligible fashion and written in standard English?

Reviewer #1: Yes

Reviewer #2: No

Reviewer #1: This was a very difficult paper to read and follow. These investigators are continuing work on samples obtained more than 10 years ago in a study of the effects in vivo and in vitro of beta-interferon, in MS patients. This is a follow-up paper from the EBioMed paper in 2019.

See attached review

Reviewer #2: The study analyzes responses to IFN-b therapy from various categories of MS in vivo and invitro to increase our understanding of why these groups respond differently to treatment. They do this by analyzing STAT1 activation and responsive genes in PBMC and sera from these different groups following in vivo and in vitro exposure to IFN-b. They propose that response differences between CR and PR may be responsible for disease state differences between these groups and reflects derangement in the type I interferon system in MS in general. While this reviewer sees value in the present work, a number of deficiencies should be addressed in the data presentation, description, and discussion as follows.

In the introduction, a complete description of IFN (types of IFNs) and STAT1 (types of STAT1-containing factors) signaling should be given to effectively explain to the reader how IFN-b potentially leads to the complex responses seen in the various MS subject groups relevant to the present study. A schematic diagram to this affect would be helpful to explain potential differences in responses to IFN-b in PR and CR to be analyzed.

The authors state that long term IFN-b therapy reverses subnormal STAT1 activation and cytokine levels presumably meaning therapy returns STAT1 activity and cytokine profiles to normal levels. However, normal levels are not shown for direct comparison and statistical analysis in this dataset. Please add these data.

In the data on pS-STAT1 in Tables 1 and 2 obtained from westerns, it is important to know whether the authors normalize this value to total STAT1 protein or not. It would be helpful to add this data to Tables 1 and 2 as these may be already available from Table 4A+B on U-STAT1.

Many of the tables for example Table 3A, could be more effectively presented to the reader in histograms (bar graphs) where statistical comparisons between groups of subjects could be displayed. Even though it is clear that the authors describe significant induction of for instance pS-STAT1 or Mx protein within a group (eg. PR) in Tables 2A and 3A, respectively), is not clear from the tables that differences in induction between PR and CR are significantly different. Please clarify this issue and consider changing the presentation of these data from table to bar graphs where the authors make a point that there is a difference between subject groups. It is suggested that Tables 4B-C also be converted in this way.

A statistical analysis of data in Tables 5 and 6 is appropriate to support statements in the text. Again, consider bar graphs indicating difference between groups.

The discussion does not highlight how the findings of this study are specifically novel. This could also be added to the abstract. In the discussion, detailed explanation of the strengths and weaknesses of these new findings and why the study is important in understanding why IFN-b works on some subject groups but not others. Finally, a schematic figure would help the reader understand the reasoning behind the differing activities of IFNs in MS.

It would be helpul in a revision to have all the figure legends moved from the results section and placed either with the individual figures and/or in a figure legend section.

**Do you want your identity to be public for this peer review?** For information about this choice, including consent withdrawal, please see our Privacy Policy

Reviewer #1: No

Reviewer #2: No

---

## [Author Response · Author response to Decision Letter 1]

22 Jul 2025

Dear Editors and Reviewers,

We agree with your comments and those of the reviewers. We truly appreciate the amount of effort and insights from the reviewers. This study is complicated but well-controlled with internal validations of molecular differences in IFN responses in subgroups of MS patients with different clinical responses.

We have corrected ambiguous statements and added importance and MS relevance throughout. We changed many tables to more easily readable bar or line graphs. We collaborated with our statistics department on the issue of multiple comparisons and study design. FYI, we have not used AI to write this paper and thus have many new insights into the immune pathology of MS.

Editor: 1. Upload as 'Response to Reviewers'

a. I have noticed that the presentation is somewhat disorganized and cluttered. I suggest including the most important figures and tables in the main manuscript and placing the remaining ones in supplemental materials.

[Revised per suggestions of editor and 2 reviewers. We have shorted the paper wherever possible. It is not repetitious but does contain a large amount of new data and ideas.]

b. Please also revise your abstract to make the sentences more specific. For example, the statement "The balance of Th1, Th2, and neurotrophic proteins was disrupted in therapy-naïve MS" requires clarification regarding the specific nature of the disruption.

[Agreed, and changed throughout abstract]

2. Data Availability Statement is currently missing [the repository name and/or the DOI/accession number of each dataset OR a direct link to access each database]. Will need de-identified raw data.

[lines 137-139 in untracked version. Gene accession # is also linked.]

6/30/2025 Responses to Redactor

2a submissions reporting blot or gel data to comply in full with the reporting requirements

[line 139: Protein raw data now in supplements S1 and S1_raw_images—line 292. We have clarified gel preparation and analysis techniques in Methods [199-204—see note for Reviewer #2]. The large number of Western gels have been converted to PDF and HIPPA-de-identified.

We have added an author who performed the very time-intensive gel conversions and labeling. ]

3. Refer to Figures 7B, 7C, 8B, 8C, 9B, 9C in your text as, if accepted, production will need this reference to link the reader to the figure.

Changed throughout. 7a, 7b... We kept 7A-E, as PLOS-published papers use this format.

4. Please include a caption for figures 1 and 2.

[92 and 293, 294 Added to text]

Reviewer #1

This was a very difficult paper to read and follow. Otherwise, the Discussion seems too long.

[Thank you for heroic efforts in reading this and for the important comments. We chose PLOSone because this is an information-rich and complex paper. We have focused on data and principles relevant to MS and tried to avoid repetition and highlight new and important concepts. ]

[This study integrates the abnormal IFN regulation that we have discovered in MS with in vivo induction by IFN-beta to probe the system, using readouts of changes in IFN-induced proteins plus cytokines and neurotrophic factors. The large number of curated targets yielded considerable amounts of data and new insights into MS pathology, therapies, and possible biomarkers. It has taken a long time to analyze the data and develop new insights. The comments help with clarity and weighting and should make it easier to read.]

This is a follow-up paper from the EBioMed paper in 2019.

[Text lines 105-110 in non-tracked version: Follow-up and comparison to gene data is emphasized more. Also included in Discussion.]

These investigators are continuing work on samples obtained more than 10 years ago in a study of the effects in vivo and in vitro of beta-interferon, in MS patients. The samples were collected in a study that ended in 2012. When were these assays done? If done recently, how were the samples maintained during the years after the study?

[144-155 Sample collection dates added.]

[216-220 Assays of sera stored at -80oC for 0 to 5 years were all run within a month of each other. We used manufacturer’s guidelines, …]

The MS patients are divided into people on beta-interferon who are stable over approx 5 years, while another group had a relapse during BIFN treatment. Some samples also taken during a relapse. The study appears to have been well-planned and carried out.

Findings were difficult to follow, as it involves tables of numerical data, which are hard to read even with color coding.

Graphic representation of the study design was greatly appreciated. An overall graphic summary of what the Investigators believe is happening, and the key differences between CR and PR would very much improve reader understanding.

Rvwer #1,#2 [92 New Fig 1 shows IFN signaling cascade and abnormalities in MS defined in this paper.]

Graphic representations of the results would be helpful too

#1,#2 [Most tables have been converted to bar graphs—Easier to follow.]

(e.g. longitudinal changes in the PR vs CR groups, including SEM bars to demonstrate overlap between the groups over time).

[117-118, 130-134 Clinical follow-up was over 5 years, and this is clarified in text. Additional longitudinal data, for instance, repeat stimulation at a given dose, is not available. Mentioned in limitations: 3…).]

Many different comparisons between IFNB use and dose with transcription factors, neurotrophins, cytokines, and chemokines were made. The Statistical handling of these multiple comparisons seems insufficient, with the use of primarily parametric t-tests or Mann-Whitney U tests. This reviewer would have expected more corrections for these multiple comparisons, or discussion of why they were not done. The potential for identifying spurious associations may have been compounded by the one-sided nature of some of the t tests, as well as the small-ish number of subjects. It is this Reviewer’s opinion that the advice of a statistician should be sought, and perhaps re-analysis of much or all of the data.

#1,#2 [Statistical input from our Biostatistics Laboratory was helpful, and supports the original conclusions. Most conclusions and comparisons held, although some p-values fell slightly with FDR correction, and are listed in the 4-digit PLOSone format. The data are internally consistent in almost all comparisons and complement published literature. We used p-values with thresholds for chance of spurious results, *5%, **1%, and ***0.1%.

[204-214 Prospective nature of targets is mentioned. Also in discussion. {The targets are not random, but were prospectively curated for MS relevance.}

[261-271 and figs. A simple Bonferroni correction is too stringent for big data. We have now used False Discovery Rate (FDR). Data are from a huge database of patient characteristics and assay readouts -- the largest of its 19 tabs is a grid with 2,084 lines and 168 columns.

We added nonparametric analyses when indicated and removed several outlier values using the interquartile range method.

[270] We added a Wald test, upon our Biostatistician’s suggestion for Figs 9-13. This analyzes cytokine groups such as pro- vs. anti –inflammatory. It reflects the serum milieu better than individual comparisons.]

In a few instances, the results in the PR and CR groups were at odds, for example MxA induction in vitro (Table 3c). Could this be spurious? Add spurious to limitations

Interpretations regarding PR versus CR might also be spurious, and not (based on these data) able to be applied to individual patients.

#1, #2 [842-868, Limitations, and multiple sites in Results and Discussion. We believe the findings are not spurious, because they match gene and protein expression in other cohorts of MS patients [Refs 3-7,14,27,47-49,66,78] including PR and CR type patients [4,14,27,66,78], plus data from Comabella and Montalban, Barcelona [49], and importantly are internally consistent within the current paper.

[349-354 The aberrant (i.e., non-significant change) response of MxA to double-dose (not single-dose) injection is mentioned, with potential mechanisms such as double-dose induction of SOCS proteins. It contrasts with serum IFN, p-S-STAT1, U-STAT1, and single dose MxA all of which parallel each other, and are greater in PR than CR.]

[The change in the many significant comparisons was 1.5- to 3-fold. Though statistically and likely biologically significant, this is too low to be used as a clinical test. Our data can be used in the future to help design multi-analyte panels and to target immune abnormalities in MS.]

In almost all datasets, there was considerable overlap between the mean+/- SEM results in the two groups. Thus, the following Conclusion: “IFNB-induced proteins and IFN levels predict clinical response to IFNB” is overly zealous. It implies an ability to predict that is not clearly present in the results.

[Agreed. “Predict” is changed to “are linked to” throughout the paper.]

Vitamin D was quantitated, but it seemed "added on" and the rationale for examining it in this context was unclear until the Discussion (but even after reading, it still seemed added on). Recent studies do not support that high dose Vit D affects MS, making this part of the manuscript perhaps of little relevance.

[96-100, 609-650 785-813 The effect of vitamin D on virus infections and IFN signaling is now in Intro and also expanded in Discussion. The lack of effect of vitamin D on patients with adequate serum levels, and those receiving glatiramer, which causes a large Th2 shift {Cassard/Mowry [76]}, is explained and contrasted to benefit with IFN-� (Th1 shift reversed by vitamin D) and fingolimod (Ascherio data [75], and 2 papers, one just published in JAMA [72,73], showing delay in onset of a 2nd demyelinating attack, especially with low baseline serum vitamin D levels.).

It was also odd that serum vitamin D correlated with serum cytokines in PR but not in CR, after IFNB injections. Could this be a spurious finding? Should this part be removed from the manuscript?

[635-650, 791-795 Mentioned in Results and Discussion. The mechanism is unexplained. But, considering different IFN responses and cytokine profiles between PR and CR, it bears study in subgroups of MS patients and in combination with other drugs.]

More of an explanation of why double dose IFNB was used is needed. This is not standard dose and why was it used in this study?

[111-116, 157-165 This was not clear. The need to overcome IFN resistance during attacks, the paired use of double doses during stable periods, and the need for a standard-dose control is now clarified. Not in paper: Clinically, dozens of patients with a history of adverse reactions to steroids report that double dose/more frequent interferon injections speed up recovery from exacerbations at least as well as steroids do [ATR].]

There were some statements in this paper that seemed stated as fact, but unproven. Here are some that seem overstated:

1. ”….. virus infections in MS are half as frequent as in healthy controls [36]” Are there other references or studies beyond Ref 36?

[94-100, 669-674 More references are provided [37-40], all from the period predating availability of disease-modifying therapy--which might impact the rate of infections.]

[786-797 Added: the connection to milder COVID with higher vitamin D levels, and with pre-COVID IFN therapy. {We are not going into the weeds to debate whether cod liver oil, high in vitamin D, is an alternative to highly-effective measles vaccinations, but the information here does have broad implications for connecting COVID responses, Vit D, and IFN signaling and immune control}]

2. Reference to “ a serum cytokine storm” during MS exacerbations. This is not a commonly held concept for MS relapses, to Reviewer’s knowledge. This should be toned down or even removed, or alternatively more data to support this statement added. The paper referenced is from the 1990’s.

[537-544 This comment is very important. The logical and historically widely-accepted idea (by me, Barry Arnason, past NIH MS researchers, and others who interacted with Hans Link) appears to be wrong. Our data argues against it, as cytokine levels were remarkably and consistently low in stable untreated MS, but dysregulated. Cytokine levels did rise with exacerbations. Mentions of the storm as dogma have been deleted.]

3. Description on Line 837 of NMO as being “an inflammatory CNS demyelinating disease” does not go far enough to describe the destructive nature of NMO. As these authors know well, NMO involves far more than inflammatory demyelination and it is likely that the demyelination is secondary.”

[837 is now 709-718 This was sloppy; it is now corrected. We also more clearly contrast our demonstration of the excessive IFN response in NMO and SLE to the low IFN response in MS.]

A paragraph devoted to Study Limitations in Discussion is needed, but missing.

[842-868 Limitations follow the section on lack of correlation between RNA and protein levels.]

Reviewer #2

The study analyzes responses to IFN-b therapy from various categories of MS in vivo and invitro to increase our understanding of why these groups respond differently to treatment. They do this by analyzing STAT1 activation and responsive genes in PBMC and sera from these different groups following in vivo and in vitro exposure to IFN-b. They propose that response differences between CR and PR may be responsible for disease state differences between these groups and reflects derangement in the type I interferon system in MS in general. While this reviewer sees value in the present work, a number of deficiencies should be addressed in the data presentation, description, and discussion as follows.

In the introduction, a complete description of IFN (types of IFNs) and STAT1 (types of STAT1-containing factors) signaling should be given to effectively explain to the reader how IFN-b potentially leads to the complex responses seen in the various MS subject groups relevant to the present study. A schematic diagram to this affect would be helpful to explain potential differences in responses to IFN-b in PR and CR to be analyzed.

#1, #2 As above, [92 Fig 1 added…]

The authors state that long term IFN-b therapy reverses subnormal STAT1 activation and cytokine levels presumably meaning therapy returns STAT1 activity and cytokine profiles to normal levels. However, normal levels are not shown for direct comparison and statistical analysis in this dataset. Please add these data.

[403-409, Results, 709-718 Serum IFN levels in NMO, SLE and treated and untreated MS are contrasted. We have measured serum IFN in MS vs. NMO vs. HC [5,6], induction of STAT and MxA proteins in different forms of MS vs. HC [3,5], gene induction in MS vs. HC [4,14,47,48,66], and protein in MS vs. HC [7,47, but none had the extensive analysis if IFN responses performed in the current paper. We did not repeat these assays in interest of time, $$, and paper length.]

In the data on pS-STAT1 in Tables 1 and 2 obtained from westerns, it is important to know whether the authors normalize this value to total STAT1 protein or not. It would be helpful to add this data to Tables 1 and 2 as these may be already available from Table 4A+B on U-STAT1.

[198-203, and 2a Revisions, above. Values are normalized to actin on Westerns. We also mention our use the same blot for sequential probing to reduce variability. To avoid variability between Westerns, we stained for p-STAT1 and actin, then stripped them and reprobed for U-STAT1 and actin on the same nitrocellulose membrane.]

[375-377 and Fig legend Normalization to U-STAT1 is an interesting concept. U- and p-STAT migrate at only slightly different rates on Westerns but are recognized by different antibodies, so we were able to calculate reliable values with the methods above. The p-STAT1/U-STAT1 ratio surprisingly did not change markedly although trends are now mentioned. Reanalysis did show that p-S-STAT1/U-STAT1 tended to be higher in CR than PR.]

Many of the tables for example Table 3A, could be more effectively presented to the reader in histograms (ba

---

## [Editor Report · Decision Letter 1]

7 Aug 2025

IFN-β therapy rescues dysregulated IFN-stimulated proteins, serum cytokines, and neurotrophic factors in multiple sclerosis: Multiplex analysis of short-term and long-term IFN responses

PONE-D-24-55183R1

Dear Dr. Feng,

We’re pleased to inform you that your manuscript has been judged scientifically suitable for publication and will be formally accepted for publication once it meets all outstanding technical requirements.

Kind regards,

Luwen Zhang

Academic Editor

PLOS ONE
---

## [Editor Report · Acceptance letter]

PONE-D-24-55183R1

PLOS ONE

Dear Dr. Feng,

I'm pleased to inform you that your manuscript has been deemed suitable for publication in PLOS ONE. Congratulations! Your manuscript is now being handed over to our production team.

Kind regards,

on behalf of

Dr Luwen Zhang

Academic Editor

PLOS ONE